# *Video-RAG*: Visually-aligned Retrieval-Augmented Long Video Comprehension

Yongdong Luo[1]  Xiawu Zheng[1]*  Guilin Li[1]  Shukang Yin  Haojia Lin[1]
Chaoyou Fu[2]  Jinfa Huang[3]  Jiayi Ji[1]  Fei Chao[1]  Jiebo Luo[3]  Rongrong Ji[1]

[1]Key Laboratory of Multimedia Trusted Perception and Efficient Computing,
Ministry of Education of China, Xiamen University, 361005, P.R. China
[2] Nanjing University  [3] University of Rochester

## Abstract

Existing large video-language models (LVLMs) struggle to comprehend long videos correctly due to limited context. To address this problem, fine-tuning long-context LVLMs and employing GPT-based agents have emerged as promising solutions. However, fine-tuning LVLMs would require extensive high-quality data and substantial GPU resources, while GPT-based agents would rely on proprietary models (e.g., GPT-4o). In this paper, we propose **Video R**etrieval-**A**ugmented **G**eneration (**Video-RAG**), a training-free and cost-effective pipeline that employs visually-aligned auxiliary texts to help facilitate cross-modality alignment while providing additional information beyond the visual content. Specifically, we leverage open-source external tools to extract visually-aligned information from pure video data (e.g., audio, optical character, and object detection), and incorporate the extracted information into an existing LVLM as auxiliary texts, alongside video frames and queries, in a plug-and-play manner. Our **Video-RAG** offers several key advantages: (i) lightweight with low computing overhead due to single-turn retrieval; (ii) easy implementation and compatibility with any LVLM; and (iii) significant, consistent performance gains across long video understanding benchmarks, including Video-MME, MLVU, and LongVideoBench. Notably, our model demonstrates superior performance over proprietary models like Gemini-1.5-Pro and GPT-4o when utilized with a 72B model. Codes are available at https://github.com/Leon1207/Video-RAG-master.

## 1 Introduction

With the advancements in Large Language Models (LLMs), numerous studies have been conducted to enhance their ability to comprehend and process videos [12, 16, 18, 51, 24, 47, 19, 3, 2, 50, 23, 25, 17], collectively termed Large Video-Language Models (LVLMs). Although current LVLMs have demonstrated promising performance in understanding short videos, effective comprehension of extremely long videos continues to be a major challenge.

To address this challenge, recent studies [49, 45, 35, 42, 55] have sought to extend the reasoning context length of LVLMs, essentially finetuning long-context LVLMs for long video understanding. LongVA [49] first introduces increasing the token capacity of an LLM and transferring its long-context comprehension capabilities to video data. However, training such a model requires pre-training on an extended corpus, and often there are distribution shifts between deployment videos and finetuning videos. As demonstrated in Video-MME [6], LongVA declines when increasing the video frame sampling rate from 128 to 384 (52.6% → 51.8%). This outcome suggests that simply increasing the

---

*Corresponding author: zhengxiawu@xmu.edu.cn

39th Conference on Neural Information Processing Systems (NeurIPS 2025).

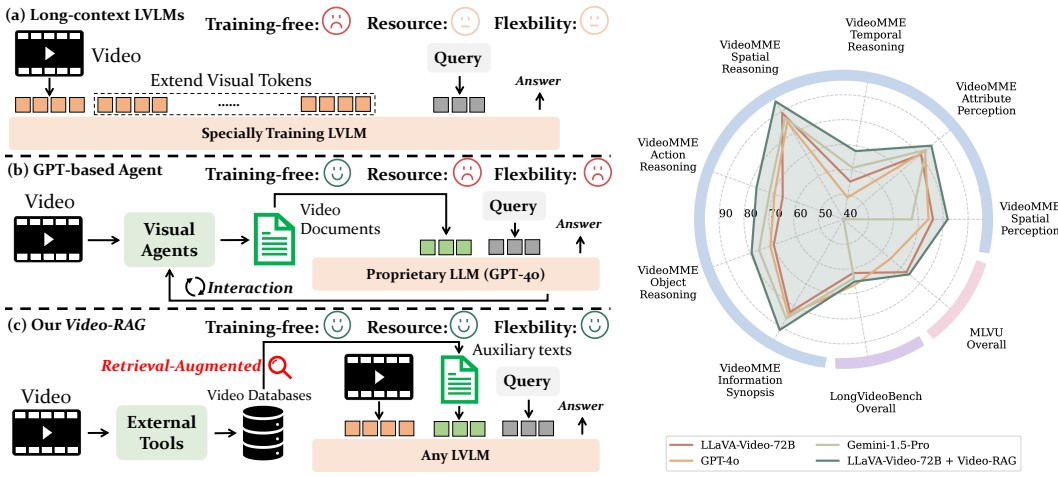

Figure 1: (Left) Two common approaches for understanding long videos, alongside our Video-RAG. Video-RAG provides a resource-efficient, training-free pipeline compatible with any LVLM. By leveraging RAG, it retrieves auxiliary texts for input, leading to notable performance enhancement. (Right) Comparison of the performance of Video-RAG with LLaVA-Video-72B [52], Gemini-1.5-Pro [32], and GPT-4o [29] across various benchmarks, including the sub-tasks from Video-MME [6] (we focus only on those that outperform Gemini-1.5-Pro), LongVideoBench [43], and MLVU [54].

number of sampled frames not only leads to information redundancy but also imposes additional challenges for the model to handle complex reasoning. Retrieval-Augmented Generation [14] (RAG) is a technique that enhances generative tasks by retrieving relevant documents from an external corpus, thus improving response quality in LLMs. Recent studies have begun exploring the integration of RAG with video-based tasks [1, 27, 48, 33], employing tools to process videos in long contexts and sending them to a proprietary model for generation, which is known as the GPT-based Agent method. However, they come with serval limitations. First, most of them process long video content as plain text, subsequently utilizing the RAG mechanisms to retrieve relevant documents for LLMs. Therefore, they lack alignment with the visual context of the video, resulting in a loss of critical visual information. Second, they are often resource-intensive in multi-turn interactions and typically require powerful LLMs to function as the driving force, thus limiting their flexibility and generative capabilities. Executing the whole Video-MME [6] using VideoAgent [4] requires approximately 20 days and incurs a substantial consumption of GPT-4o API tokens.

In this study, we propose Video-RAG, an effective RAG pipeline that can be seamlessly integrated with any LVLM. Specifically, instead of simply increasing the number of sampled video frames, we propose to replace the corresponding extended visual tokens with auxiliary texts extracted from pure video data by invoking open-source foundation models, such as optical character recognition (OCR), automatic speech recognition (ASR), and object detection. These auxiliary texts are more aligned with the visual context while providing additional information beyond the visual data, as demonstrated in [20, 4]. Besides dealing with the context windows limit of LVLMs, we employ RAG in Video-RAG to filter auxiliary texts, ensuring their relevance to the user's query in the text embedding space. As sampled visual context often lacks explicit alignment with the instructions, the auxiliary texts can facilitate cross-modality alignment while reducing the modality divide. As illustrated in Figure 5, with Video-RAG, the retrieved auxiliary texts help guide the LVLM to pay more attention to the query-relevant keyframes, while simultaneously facilitating cross-modality alignment between query and keyframes. In this framework, an LVLM serves as the central component of Video-RAG, processing visual tokens to preserve detailed visual context and minimize potential information loss. Moreover, the retrieval process is parallelly executed in a single operation, ensuring efficiency.

We evaluate Video-RAG across several long video benchmarks, including Video-MME [6], MLVU [54], and LongVideoBench [43]. By applying the Video-RAG to seven distinctive open-source LVLMs, we achieve an average performance improvement of 2.8% on Video-MME with only 2.0K text tokens addition (equal to 14 frames in most configuration) per case, while beating the proprietary LVLM when integrated with the 72B model, as shown in the right part of Figure 1. Applying Video-

RAG to a 7B LVLM only requires an additional 8GB of inference GPU memory and approximately 5 seconds of inference time per case (details in the Supplemental Material).

In summary, our contributions are as follows:

- **We integrate RAG into open-source LVLMs:** Video-RAG incorporates three types of visually-aligned auxiliary texts (OCR, ASR, and object detection) processed by external tools and retrieved via RAG, enhancing the LVLM. It's implemented using completely open-source tools, without the need for any commercial APIs.

- **We design a versatile plug-and-play RAG-based pipeline for any LVLM:** Video-RAG offers a training-free solution for a wide range of LVLMs in a plug-and-play manner, delivering performance improvements with minimal additional resource requirements.

- **We achieve proprietary-level performance with open-source models:** Applying Video-RAG to a 72B open-source model yields proprietary-level performance, surpassing models such as GPT-4o and Gemini-1.5-Pro.

## 2  Related Work

### 2.1  Large Video-Language Models

With the rapid advancement of large language models (LLMs), there has been increasing interest in developing generalist video models capable of handling video-related tasks. Video-ChatGPT [28] extracts features from individual frames and aggregates them through both spatial and temporal pooling operations. VideoChat [16] encodes videos by generating both textual descriptions and video appearance embeddings. Video-LLaVA [18] aligning image and video encoders during a pre-processing phase, using a shared projector to map the encoded representations into a common language space. LLaVA-NeXT-Video [51] extends LLaVA-NeXT [22] by fine-tuning the model specifically on video data. Despite their contributions, these approaches face challenges when processing long videos, primarily due to the limited number of frames sampled for analysis.

### 2.2  Long-context Large Video-Language Models

Recent approaches have sought to expand the context window size to enhance long video understanding. LongVA [49] and Long-LLaVA [45] address this by continuously training LLMs on extended textual data, to transfer their long-text comprehension capabilities to video processing. INTP [35] introduces a video token rearrangement technique while proposing a training-free method for extending the LLM context window, allowing LVLMs to process increased visual tokens. However, these methods face challenges in striking a balance between the high computational costs associated with sampling video frames and the limited performance improvements achieved. Due to the inherent redundancy in video content and constraints on model capacity, performance degradation may occur when the number of sampled frames surpasses a certain threshold.

### 2.3  GPT-based Agent Video Understanding

Initial efforts [46, 41, 8, 38, 33] have employed LLMs to interact with tools to process visual information as structured long context for question-answering. MM-VID [20] enhances long video understanding by aligning video frames with corresponding text descriptions. VLog [21] leverages multimodel pre-trained models to capture and interpret visual and audio information, summarizing it into documents for video comprehension. VideoAgent [4], DrVideo [27], and OmAgent [48] integrate multimodal inputs and enable dynamic querying of video segments to support long video reasoning tasks. VideoRAG [33] and VideoRAG [11] achieve a tighter integration between the RAG framework and proprietary models. However, these methods take a long time to process videos while relying on proprietary models (e.g., GPT-4o), thus limiting their efficiency and adaptability to other open-source frameworks.

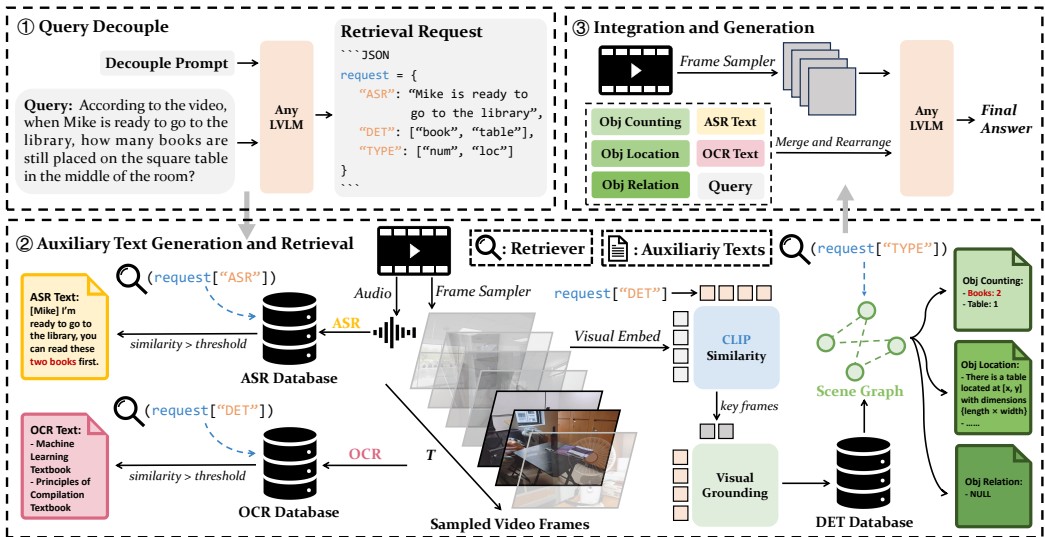

Figure 2: The framework of our Video-RAG. In the query decouple phase, the LVLM is prompted to generate a retrieval request for auxiliary texts. Next, in the auxiliary text generation and retrieval phase, the video is processed **in parallel** to extract three types of textual information (OCR, ASR, and object detection), and the relevant text is retrieved as the auxiliary text. Finally, in the integration and generation phase, auxiliary texts are combined with the query and the video to generate the response.

## 3 Method

We propose a novel, training-free pipeline for large video-language models (LVLMs), named Video-RAG, which can be integrated into any LVLM. As illustrated in Figure 2, our pipeline comprises three key phases: **(i) Query Decouple:** In this phase, the user's query is decomposed into a retrieval request aimed at extracting auxiliary texts from the target video. **(ii) Auxiliary Text Generation & Retrieval:** Multiple auxiliary texts are generated from the queried video in parallel. Then, the retrieval request is used to obtain relevant external information. **(iii) Integration and Generation:** This phase integrates the retrieved auxiliary texts with the user's query, feeding this combined input into the LVLMs to generate the final response.

### 3.1 Large Video-Language Model

Given a video $\mathbf{V}$, a frame sampler first sample $N$ frames $\mathbf{F}$. Most existing methods uniformly sample frames from a video for both effectiveness and simplicity. Then, video features are extracted as $\mathbf{F_v} = \texttt{VisualEnc}(\mathbf{F})$, where $\texttt{VisualEnc}$ is an image-based visual encoder, such as CLIP-L [30]. Finally, the video features $\mathbf{F_v}$ and the user's query $\mathbf{Q}$ are fed into the LVLM to generate an output $\mathbf{O}$:

$$\mathbf{O} = \texttt{LVLM}(\mathbf{F_v}, \mathbf{Q}) \tag{1}$$

### 3.2 Query Decouple

In this phase, upon receiving a user's query about the video, the LVLM begins by decoupling the query and generating retrieval requests, denoted as $\mathbf{R}$, for auxiliary texts. During this phase, the LVLM processes only textual information, without access to video frames, and the output requests are formatted in JSON. We prompt the LVLM using a decoupling prompt $\mathbf{P}$ to generate the following retrieval requests as necessary: (i) $\mathbf{R}_{asr}$: Requests about automatic speech recognition, to extract audio information from the video that may pertain to the query. (ii) $\mathbf{R}_{det}$: Requests for identifying physical entities within the video that may assist in answering the query. (iii) $\mathbf{R}_{type}$: Requests for details about the location, quantity, and relationships of the identified physical entities. These requests, which may be NULL (the corresponding information is not required), are then parsed and forwarded to the auxiliary text retrieval phase. The entire process can be described as:

$$\mathbf{R} = \texttt{LVLM}(\mathbf{P}, \mathbf{Q}), \ \ \mathbf{R} = \{\mathbf{R}_{asr}, \mathbf{R}_{det}, \mathbf{R}_{type}\} \tag{2}$$

### 3.3 Auxiliary Text Generation

In this phase, we first generate the auxiliary texts from the video and then retrieve them to assist the LVLMs according to the retrieval requests $\mathbf{R}$. As the length of the video increases, the number of tokens generated from the processed data also grows, leading to an increase in redundant information. Additionally, current open-source models are constrained by the limited length of their context windows, which may prevent them from fully processing all auxiliary texts. To address this issue, we draw inspiration from Retrieval-Augmented Generation (RAG) [14], retrieving only the auxiliary texts relevant to the user's query. Before retrieval, we construct the necessary databases from the given video in parallel. Specifically, we implement three distinct databases: the Optical Character Recognition (OCR) database, denoted as $DB_{ocr}$; the Automatic Speech Recognition (ASR) database, denoted as $DB_{asr}$; and the Object Detection (DET) database, denoted as $DB_{det}$.

**OCR database.** Current LVLM are still illusory in their ability to accurately recognize characters, and their performance often falls short compared to proprietary models. To better leverage the information contained in video frames and reduce hallucinations, we employ a proprietary OCR model to extract text from each video frame with the same frame-sampled strategy as LVLMs. Specifically, we use EasyOCR [10] as our text recognition model and segmented the recognized texts on a per-frame basis, denoted as $\mathbf{T}_{ocr}$. Subsequently, we implemented RAG by utilizing the advanced text encoding model Contriever [9] to encode the fetched OCR texts into text embeddings $\mathbf{E}_{ocr}$. These embeddings are then stored in a database with the FAISS index [13], a library designed for efficient similarity search and clustering of dense vectors. The entire building process can be formally described as:

$$\mathbf{T}_{ocr} = \texttt{EasyOCR}(\mathbf{F}) \tag{3}$$

$$DB_{ocr} \xleftarrow{\texttt{FAISS}} \mathbf{E}_{ocr} = \texttt{Contriever}(\mathbf{T}_{ocr}) \tag{4}$$

**ASR database.** Audio information (e.g., subtitles) plays a crucial role in video comprehension, often providing additional context that may not be available through visual cues alone. To incorporate them, we first extract the raw audio $\mathbf{U}$ from the video and then transcribe them into texts $\mathbf{T}_{asr}$. Specifically, we use Whisper [31] as our audio transcription model. Since the recognized texts can be quite extensive, we chunk and encode them into a vector database, following the same procedure used to construct the OCR database. The building process can be formally described as:

$$\mathbf{T}_{asr} = \texttt{Whisper}(\mathbf{U}) \tag{5}$$

$$DB_{asr} \xleftarrow{\texttt{FAISS}} \mathbf{E}_{asr} = \texttt{Contriever}(\mathbf{T}_{asr}) \tag{6}$$

**DET database.** While LVLMs demonstrate strong performance in object recognition, they continue to face challenges such as object counting, precise object localization, and understanding relative relationships between objects. To mitigate the issue of hallucination, which can stem from these challenges, we incorporate object detection information as auxiliary texts. We leverage a visual grounding model to extract both the object categories and their corresponding positions from sampled video frames. This approach helps provide more accurate and context-aware object detection. To enhance processing efficiency, we limit object detection to keyframes only. Specifically, we compute the CLIP similarity [30] between the object retrieval request $\mathbf{R}_{det}$ and the sampled video frames $\mathbf{F}$ and select relevant keyframes $\mathbf{F}_{key}$ based on a threshold $t$:

$$\mathbf{F}_{key} = \texttt{CLIP\_similarity}(\mathbf{R}_{det}, \mathbf{F}) > t \tag{7}$$

Once the keyframes are identified, we utilize APE [36], an efficient open-vocabulary object detection model that accepts object descriptions as prompts to detect relevant objects within frames based on retrieval queries. The capability of APE makes it particularly well-suited to our requirements for on-demand object retrieval. Finally, the detected objects' categories and their corresponding positional information are stored in the DET database using natural language representations:

$$DB_{det} \leftarrow \mathbf{T}_{det} = \texttt{APE}(\mathbf{F}_{key}, \mathbf{R}_{det}) \tag{8}$$

## 3.4 Auxiliary Text Retrieval

During the retrieve phase, we employ Contriever [9] to encode the user's query and the parsed requests for OCR and ASR into text embeddings, then concatenating to form the final query request $\mathbf{E}_{req} = \texttt{Contriever}(\texttt{Concat}(\mathbf{R}, \mathbf{Q}))$, $\mathbf{R} \in \{\mathbf{R}_{ocr}, \mathbf{R}_{asr}\}$. Then we retrieve the auxiliary texts from $DB \in \{DB_{ocr}, DB_{asr}\}$ by the FAISS tool, which computes the vector similarity between the query and text chunks stored in the database. Text chunks with a FAISS similarity score greater than threshold $t$ are indexed as the retrieval results $\mathbf{A} \in \{\mathbf{A}_{ocr}, \mathbf{A}_{asr}\}$. The process can be formulated as:

$$\mathbf{A} \xleftarrow{\texttt{Index}} \texttt{FAISS\_similarity}(DB, \mathbf{E}_{req}) > t \tag{9}$$

Since the text generated by the detection model is in a raw format ("category: [x_min, y_min, length, width]"), it challenges LVLMs to understand the relative relationships between objects. We preprocess the object information using a scene graph, which helps to represent spatial and relational information more explicitly. This preprocessing allows us to construct more coherent and semantically meaningful texts, denoted as $\mathbf{A}_{det}^p$, which are more readily interpretable by LVLMs. We incorporate three types of object information for each video keyframe: **(i) Object Location $\mathbf{A}_{loc}$:** This refines the positional information of the object, formatted as: "Object {node ID} is a {object category} located at coordinates [x, y] with dimensions {length $\times$ width}" **(ii) Object Counting $\mathbf{A}_{cnt}$:** This counts the number of objects and generates text in the following format: "Object counting: - {object category}: {number}" **(iii) Relative Positional Relationships $\mathbf{A}_{rel}$:** This captures the relative spatial relationships between objects using the format: "Object {node ID} ({object category}) is <positional description> Object {node ID} ({object category})". By combining this information, we construct a detailed representation of the objects in the frame, denoted as $\mathbf{A}_{det}^p = \{\mathbf{A}_{loc}, \mathbf{A}_{cnt}, \mathbf{A}_{rel}\}$:

$$\mathbf{A}_{det}^p = \texttt{SceneGraph}(DB_{det}) \tag{10}$$

Finally, we acquire the object auxiliary texts based on the object information type retrieval requests $\mathbf{R}_{type}$, which selects and finalizes the object auxiliary information $\mathbf{A}_{det}$. $\mathbf{A}_{det}$ is one of the elements of the power set $\mathcal{P}$ of $\mathbf{A}_{det}^p$ selected by $\mathbf{R}_{type}$, and the retrieve process can be formulated as:

$$\mathbf{A}_{det} = \mathbf{R}_{type}(\mathcal{P}(\mathbf{A}_{det}^p)) \in \mathcal{P}(\mathbf{A}_{det}^p) \tag{11}$$

## 3.5 Integration and Generation

After obtaining different types of auxiliary texts, we organize them chronologically using natural language to create a unified auxiliary input, denoted as $\mathbf{A}_m = \texttt{Concat}(\mathbf{A}_{ocr}, \mathbf{A}_{asr}, \mathbf{A}_{det})$. These merged auxiliary inputs, along with the user's query and the sampled video frames, are then fed into the LVLM to produce the final result. The overall process can be formulated as:

$$\mathbf{O} = \texttt{LVLM}(\mathbf{F}_v, \texttt{Concat}(\mathbf{A}_m, \mathbf{Q})) \tag{12}$$

# 4 Experiments

## 4.1 Datasets

**Video-MME** [6] is a widely used benchmark for assessing the ability of LVLMs to handle detailed videos in real-world scenarios. It is divided into three subsets based on video length, with durations ranging from 11 seconds to 1 hour. **MLVU** [54] is a long video understanding benchmark with a large wide of 9 distinct tasks. It is created based on long videos of diversified lengths, ranging from 3 minutes to 2 hours with about 12 minutes average video length. **LongVideoBench** [43] is a benchmark designed to accurately retrieve and reason over detailed multimodal information from long videos, with 6,678 human-annotated multiple-choice questions in 17 fine-grained categories.

## 4.2 Implementation Details

We performed all experiments on NVIDIA A100 80G GPUs. During the auxiliary text generation phase, we first restrict the detection requests $\mathbf{R}_{det}$ generated by LVLMs in decouple prompt then

Table 1: Performance on the Video-MME [6] benchmark in without subtitles (w/o S), with subtitles (w/ S) and equipped with our Video-RAG (Ours), **Frames** and **Gain** means the input frame number and performance gain by applying Video-RAG compared to the baseline with subtitles. By applying Video-RAG to seven LVLMs, we observed an average performance improvement of 2.8% by adding only an average of ∼2.0K auxiliary texts compared to ∼3.0K full-subtitled tokens per sample. In particular, we perform better when applying Video-RAG with 72B LLaVA-Video [52] than the proprietary method GPT-4o [29] (77.4% vs. 77.2%). All results are our republication.

| Model | Params | Frames | Short | | | Medium | | | Long | | | Overall | | | Gain |
|---|---|---|---|---|---|---|---|---|---|---|---|---|---|---|---|
| | | | w/o S | w/ S | Ours | w/o S | w/ S | Ours | w/o S | w/ S | Ours | w/o S | w/ S | Ours | |
| *Proprietary LVLMs* | | | | | | | | | | | | | | | |
| GPT-4o [29] | - | 384 | 80.0 | 82.8 | - | 70.3 | 76.6 | - | 65.3 | 72.1 | - | 71.9 | 77.2 | - | - |
| Gemini-1.5-Pro [32] | - | 0.5 fps | 81.7 | 84.5 | - | 74.3 | 81.0 | - | 67.4 | 77.4 | - | 75.0 | 81.3 | - | - |
| *Open-Source LVLMs* | | | | | | | | | | | | | | | |
| Video-LLaVA [18] | 7B | 8 | 45.3 | 46.1 | 49.5 | 38.0 | 40.7 | 43.0 | 36.2 | 38.1 | 42.5 | 39.9 | 41.6 | 45.0 | +3.4 |
| LLaVA-NeXT-Video [51] | 7B | 16 | 49.4 | 51.8 | 56.6 | 43.0 | 46.4 | 47.4 | 36.7 | 44.9 | 46.0 | 43.0 | 47.7 | 50.0 | +2.3 |
| VITA-1.5 [7] | 7B | 16 | 67.0 | 69.9 | 71.0 | 54.2 | 55.7 | 55.4 | 47.1 | 50.4 | 52.4 | 56.1 | 58.7 | 59.6 | +0.9 |
| LongVA [49] | 7B | 128 | 61.1 | 61.2 | 66.1 | 50.4 | 53.8 | 60.4 | 46.2 | 52.9 | 59.4 | 52.6 | 56.0 | 62.0 | +6.0 |
| Long-LLaVA [45] | 7B | 64 | 61.9 | 62.4 | 67.1 | 51.4 | 56.2 | 60.4 | 45.4 | 54.7 | 60.1 | 52.9 | 57.8 | 62.6 | +4.8 |
| Qwen2-VL [40] | 72B | 32 | 75.0 | 76.7 | 77.4 | 63.3 | 69.9 | 70.2 | 56.3 | 69.2 | 71.0 | 64.9 | 71.9 | 72.9 | +1.0 |
| LLaVA-Video [52] | 72B | 64 | 80.7 | 81.8 | 82.8 | 68.7 | 73.8 | 76.3 | 62.1 | 72.2 | 73.1 | 70.3 | 75.9 | 77.4 | +1.5 |

further filter them using spaCy, ensuring they correspond to CLIP-sensitive physical entities, avoiding the inclusion of abstract concepts. In the auxiliary text retrieval phase, we set both the CLIP similarity threshold and the FAISS similarity threshold $t$ to 0.3. We employ the IndexFlatIP as the similarity calculating method of FAISS [13]. Note that we don't include the GPT-based Agent methods for comparison due to their resource-intensive nature (complete execution of Video-MME [6] costs around $2000 for API purchasing when using VideoAgent [4]). Still, we include a mini-experiment of VideoAgent in the Supplemental Material that compares the overall performance, inference time, and GPU requirements with two common long-context LVLMs and our Video-RAG.

Table 2: The overall performance in the multiple-choice task of the MLVU [54] benchmark. * donates the results of our replication.

| Model | #Params | Frames | Overall |
|---|---|---|---|
| *Proprietary LVLMs* | | | |
| GPT-4o [29] | - | 0.5 fps | 64.6 |
| *Open-Source LVLMs* | | | |
| VITA-1.5 [7] | 7B | 16 | 60.4 |
| Video-CCAM [5] | 14B | 96 | 63.1 |
| Video-XL [37] | 7B | 256 | 64.9 |
| Aria [15] | 25.3B | 256 | 70.6 |
| LLaVA-Video* [52] | 7B | 64 | 70.8 |
| Oryx-1.5 [26] | 32B | 128 | 72.3 |
| LLaVA-Video* [52] | 72B | 64 | 73.1 |
| LLaVA-Video + Video-RAG | 7B | 64 | 72.4 |
| LLaVA-Video + Video-RAG | 72B | 64 | **73.8** |

Table 3: The overall performance on the validation set of LongVideoBench [43]. * donates the results of our replication.

| Model | #Params | Frames | Overall |
|---|---|---|---|
| *Proprietary LVLMs* | | | |
| Gemini-1.5-Pro [32] | - | 256 | 64.0 |
| GPT-4o [29] | - | 256 | **66.7** |
| *Open-Source LVLMs* | | | |
| VideoChat2-Mistral [16] | 7B | 8 | 39.3 |
| ShareGPT4Video [2] | 7B | 8 | 39.7 |
| LLaVA-Next-Mistral [22] | 7B | 8 | 49.1 |
| PLLaVA [44] | 34B | 16 | 53.2 |
| VITA-1.5 [7] | 7B | 16 | 53.6 |
| LLaVA-Video* [52] | 7B | 64 | 56.6 |
| LLaVA-Video* [52] | 72B | 64 | 61.9 |
| LLaVA-Video + Video-RAG | 7B | 64 | 58.7 |
| LLaVA-Video + Video-RAG | 72B | 64 | 65.4 |

## 4.3 Main Results

**Video-MME.** We evaluate our Video-RAG in five 7B open-source LVLMs, including Video-LLaVA [18], LLaVA-NeXT-Video [51], LongVA [49], Long-LLaVA [45], and two 72B LVLM Qwen2-VL [40] and LLaVA-Video [52]. Constraining by computational resources, we evaluate the LVLMs with their official frame rate setting in Video-MME except for 72B Qwen2-VL, which requires about 3K GPU memory with 768 video frame input (∼38 A100 GPUs). Results are shown in Table 1. Specifically, after applying our Video-RAG in 72B LLaVA-Video [52], we perform better than the proprietary model GPT-4o [29] (77.4% vs. 77.2%). Across the seven LVLMs used in our experiments, we gained an average performance boost of 2.8% compared to results with subtitles, especially a significant gain on long videos, demonstrating its effectiveness. This performance improvement is

achieved by incorporating token counts from approximately 14 additional video frames (equivalent to 2.0K tokens), each contributing around 144 tokens under most LVLM configurations. We obtain such a large performance enhancement because most LVLMs are pre-trained primarily within the text space and aligned with visual information, often lacking explicit alignment between embedding spaces. Auxiliary texts can serve as semantic supplements sensitive to LVLMs, facilitating model activation and easing the understanding of complex videos.

**MLVU**. We evaluate Video-RAG when integrating into the 7B and 72B LLaVA-Video [52] of MLVU [54], a benchmark that is close to performance saturation. As shown in Table 4, Video-RAG's 1.6% improvement at 7B-scale is substantial, considering that it outperforms the 32B Qryx-1.5 [26] by 0.1%, while recent 7B-scale models average only a 1.3% gain (across 15 approaches in MLVU's leaderboard). Additionally, the 72B LLaVA-Video also has a performance gain of 0.7%, which sets a new state-of-the-art.

**LongVideoBench**. We evaluate Video-RAG when applied in the 7B and 72B LLaVA-Video [52] of LongVideoBench [43]. We omit the interleaved input format introduced in LongVideoBench when applying Video-RAG. The evaluation results in Table 5 demonstrate that 72B LLaVA-Video with our Video-RAG achieves an overall performance of 65.4% on the validation set. This result surpasses the proprietary LVLM Gemini-1.5-Pro [32] by 1.4%, securing the second place, just 1.3% behind GPT-4o [29]. Meanwhile, the 7B LLaVA-Video also has a performance enhancement of 2.1% when equipped with our Video-RAG.

## 4.4 Ablation Studies

**Effect of different sampling frame number.** To explore the effect of the number of sampling frames on Video-RAG, we experience sampling frames number of 8, 16, 32, 64, 128, and 256 in 7B model LongVA [49], results are shown in Figure 3. As demonstrated, Video-RAG consistently delivers performance improvements across all frame rates, especially in long videos. The experimental results also indicate that Video-RAG can achieve higher performance gains with fewer frames, demonstrating its potential for applications under resource-constrained conditions.

**Effect of different components of Video-RAG.** To explore the effectiveness of auxiliary texts, we add DET, OCR, and ASR as auxiliary texts before and after retrieving by the RAG to evaluate Long-LLaVA-7B [45] with 32-frame setting in the Video-MME [6] benchmark. As shown in Table 4, the performance of Long-LLaVA progressively improves as auxiliary texts after retrieving by the RAG system are incrementally added (52.0% → 52.9% → 55.7% → 62.1%). Among these components, ASR auxiliary texts contribute to a general improvement for different video durations, especially for long videos. When all components are integrated, we obtain an optimal performance, as shown in

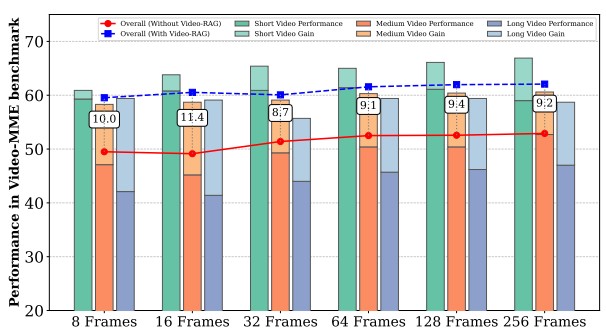

Figure 3: Performance gain with different sampling frames rate on Video-MME [6] when implement LongVA-7B [45].

the last row of Table 4. Meanwhile, the experiment shows a 2.3% improvement (59.8% vs 62.1%) in performance after incorporating RAG for retrieval, demonstrating that auxiliary texts after retrieving by the RAG system are query-aligned, which helps cross-modality alignment. We also evaluate across sub-tasks within Video-MME [6] and other video benchmarks like MLVU [54], LongVideoBench[43], and VNBench [53], more details are shown in the Supplemental Material.

**Effect of different thresholds of RAG processing.** When retrieving, we specify a similarity threshold $t$ as a criterion for information selection. In the retrieval for OCR and ASR texts, information is selected if its FAISS similarity exceeds $t$. For object detection, frames are selected as keyframes based on their CLIP similarity surpassing $t$, and the relevant information is then extracted. Setting $t$ too high may hinder the retrieval of relevant information while setting it too low can result in information redundancy and increased reasoning complexity. To investigate this trade-off, we conduct ablation experiments to evaluate the impact of different threshold values. The results are shown in Table 5.

Table 4: Results on combinations of different auxiliary texts in Video-MME [6] when using Long-LLaVA-7B [45] as the LVLM.

| RAG | DET | OCR | ASR | Short | Medium | Long | Overall |
|---|---|---|---|---|---|---|---|
| | | | | 60.3 | 51.4 | 44.1 | 52.0 |
| | | | ✓ | 62.2 | 55.4 | 54.4 | 57.4 |
| | | ✓ | ✓ | 64.0 | 56.2 | 55.0 | 58.4 |
| | ✓ | | ✓ | 63.0 | 57.3 | **56.4** | 58.9 |
| | ✓ | ✓ | ✓ | **64.3** | **58.8** | 56.3 | **59.8** |
| ✓ | ✓ | | | 61.4 | 51.9 | 45.2 | 52.9 |
| ✓ | | ✓ | | 63.2 | 53.2 | 46.3 | 54.3 |
| ✓ | | | ✓ | 65.1 | 59.1 | **60.7** | 61.6 |
| ✓ | ✓ | ✓ | | 64.1 | 54.6 | 48.4 | 55.7 |
| ✓ | ✓ | | ✓ | 64.9 | 59.0 | **60.7** | 61.5 |
| ✓ | | ✓ | ✓ | 66.3 | **60.3** | 59.3 | 62.0 |
| ✓ | ✓ | ✓ | ✓ | **66.4** | 60.2 | 59.8 | **62.1** |

Table 5: Performance with different thresholds of retrieval on Video-MME [6] when using Long-LLaVA-7B [45] as the LVLM. **#Token** and **Time** denote the total token number of the auxiliary texts and the average inference time per question, respectively.

| $t$ | #Token | Time | Short | Medium | Long | Overall |
|---|---|---|---|---|---|---|
| 0.0 | 3.6K | 36s | **67.6** | 59.4 | 59.1 | 62.0 |
| 0.1 | 3.4K | 30s | 67.0 | 59.7 | 59.1 | 61.9 |
| 0.2 | 2.7K | 18s | 66.0 | **60.2** | 59.2 | 61.8 |
| 0.3 | 1.9K | 11s | 66.4 | **60.2** | 59.8 | **62.1** |
| 0.4 | 0.8K | 8s | 65.6 | 58.0 | 58.3 | 60.6 |
| 0.5 | 0.3K | 7s | 63.1 | 54.9 | 50.2 | 56.1 |
| 1.0 | 0.0K | 6s | 60.3 | 51.4 | 44.1 | 52.0 |
| rnd | 1.9K | 11s | 65.7 | 55.8 | 56.0 | 59.1 |

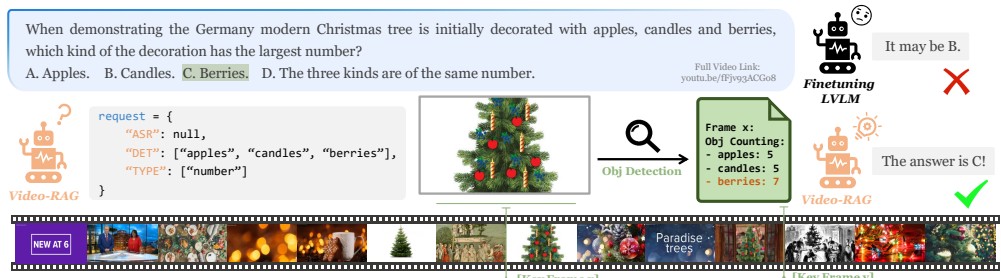

Figure 4: Qualitative result on Video-MME [6] when applying Video-RAG with LLaVA-Video [52].

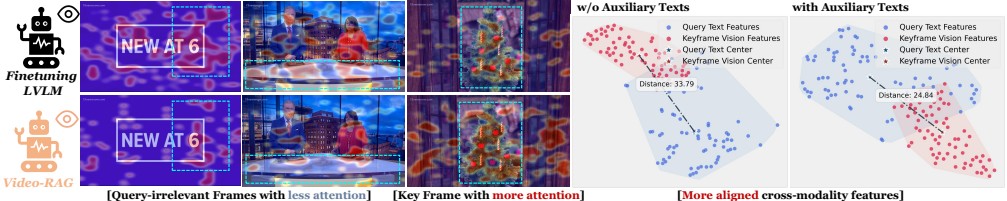

Figure 5: Grad-CAM visualizations of the last hidden state heatmap along with t-SNE visualizations of the user's query and keyframe features of the example shown in Figure 4. The retrieved auxiliary texts help **cross-modality alignment** by assisting the model to **pay more attention to query-relevant keyframes** and thus generate more robust and accurate answers to the user's query.

Notably, $t = 0$ and $t = 1$ correspond to all auxiliary texts input into the model and no auxiliary texts input, respectively. To balance performance with information density and processing time (especially APE [36] detection in keyframes), we selected a threshold of 0.3 for our implementation. More details about similarity scores are shown in the Supplemental Material. Under this configuration, the additional text length of approximately 1.9K tokens typically remains within the context window limits of open-source LVLMs. For models with more stringent context window limitations, a threshold of 0.4 may also be a viable option. We also randomly sample an equivalent token number of auxiliary texts to serve as inputs for assessing the effectiveness of RAG retrieval, as shown in the last row of Table 5.

## 4.5 Qualitative Evaluation

We present qualitative results in the case of Video-MME [6] in Figure 4 and Figure 5. As illustrated, augmenting LLaVA-Video with external tools to process and retrieve auxiliary texts from videos significantly enhances its ability to reduce visual hallucinations, thereby enabling more accurate responses to user queries. Grad-CAM [34] and t-SNE [39] visualization results also show that applying Video-RAG helps the LVLM's cross-modality alignment.

## 5 Conclusion

In this paper, we present Video-RAG for effective long video understanding through integrating retrieved auxiliary texts with LVLMs, achieving proprietary-level performance with 72B open-source LVLM. Unlike traditional methods that are resource-intensive with limited gains, Video-RAG offers a resource-efficient, plug-and-play solution leveraging only open-source tools to extract visually-aligned auxiliary texts from video data. However, Video-RAG may be limited by the visual tools we choose and their performance, which lacks adaptation. In the future, we will explore how to more efficiently integrate auxiliary texts and provide an adaptive frame selection strategy for LVLMs.

## 6 Acknowledge

This work was supported by the National Science Fund for Distinguished Young Scholars (No.62025603), the National Natural Science Foundation of China (No. U21B2037, No. U22B2051, No. U23A20383, No. U21A20472, No. 62176222, No. 62176223, No. 62176226, No. 62072386, No. 62072387, No. 62072389, No. 62002305 and No. 62272401), the Natural Science Foundation of Fujian Province of China (No. 2021J06003, No.2022J06001), and the Fundamental Research Funds for the Central Universities.

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

# Supplemental Material

## A    Decouple Query

In the initial phase of the proposed Video-RAG, we employ a decouple prompt, denoted as $\mathbf{P}$, to guide the LVLM in generating retrieval requests. In this section, we present one example of a prompt designed for multiple-choice questions, as illustrated in Figure 8.

## B    Sub-set of Video-MME

As outlined in the implementation details, we randomly sampled a subset of the Video-MME [6] dataset to evaluate a computationally resource-intensive, agent-based method with long-context LVLMs. Specifically, we selected 10% of the full dataset, comprising 30 short, 30 medium-length, and 30 long videos. Each video contains three multiple-choice questions. Importantly, we ensured that the performance ranking of the methods on the subset mirrored that of the full dataset. As shown in Tables 6 and 7, we evaluated four distinct 7B models Chat-Univi-v1.5 [12], LLaVA-NeXT-Video [51], LongVA [49], and Long-LLaVA [45] using a frame sampling rate of 16 for both the subset and the full set. Our results indicate that the performance rankings remained consistent across both evaluations.

Table 6:  Performance of Video-MME sub-set.

| Method | Short | Medium | Long | Overall |
|---|---|---|---|---|
| Chat-Univi-v1.5 [12] | 50.0 | 33.3 | 17.8 | 33.7 |
| LLaVA-NeXT-Video [51] | 54.4 | 33.3 | 23.3 | 37.0 |
| LongVA [49] | 56.7 | 50.0 | 38.9 | 48.5 |
| Long-LLaVA [45] | 58.9 | 52.2 | 40.0 | 50.4 |

Table 7:  Performance of Video-MME full-set.

| Method | Short | Medium | Long | Overall |
|---|---|---|---|---|
| Chat-Univi-v1.5 [12] | 45.7 | 39.0 | 35.7 | 40.1 |
| LLaVA-NeXT-Video [51] | 51.1 | 41.8 | 36.8 | 43.2 |
| LongVA [49] | 60.8 | 45.2 | 41.4 | 49.1 |
| Long-LLaVA [45] | 59.3 | 49.3 | 44.4 | 51.0 |

## C    Results on Video-MME Sub-Set

We examine Video-RAG against two representative methods in terms of inference time, GPU resource requirements, and overall performance. Given that GPT-based Agent methods are resource-intensive, we randomly sampled a sub-set of the Video-MME [6] for evaluation, as described in Section B. As demonstrated in Figure 6, VideoAgent [4], a typically GPT-based Agent method, requires significant time to process video and deliver suboptimal performance. Meanwhile, LongVA [49], a representative long-context LVLM, shows limited improvement from increasing the frame rate and even experiences performance degradation. Integrating our Video-RAG into the 16-frame LongVA results in substantial performance improvements while reducing GPU resource consumption. Specifically, with only increasing 8GB GPU memory compared to the base (16-frames LongVA), we achieve 11.5% overall performance improvement, while outperforming another long-context LVLM Long-LLaVA-7B [45] in 16-frames setting by 9.6% with less GPU memory requirements and compatible total inference time. These results demonstrated that our Video-RAG is lightweight with lower computing overhead than the other typical methods. Moreover, we provide detailed time consuming to construct three types of databases (which can be built in parallel) and inference per query, as shown in Table 8.

Table 8: Overall performance, databases construct and average inference time (include building databases) per query (#Time) in Video-MME-mini.

| Model | ASR | OCR | DET | Total Time | w/o subs | | w/ Video-RAG | |
|---|---|---|---|---|---|---|---|---|
| | | | | | #Time | Overall | #Time | Overall |
| VideoAgent | - | - | - | - | 14min | 47.7 | - | - |
| LongVA-16fs | 21min | 2min | 3min | max(21, 2, 3)=21min | 1s | 48.5 | 1s + 5s | 60.0 |
| LongVA-128fs | 21min | 16min | 16min | max(21, 16, 16)=21min | 8s | 54.1 | 8s + 5s | 63.3 |
| LongVA-384fs | 42min | 48min | 24min | max(42, 48, 24)=48min | 20s | 53.7 | 20s + 11s | 63.6 |

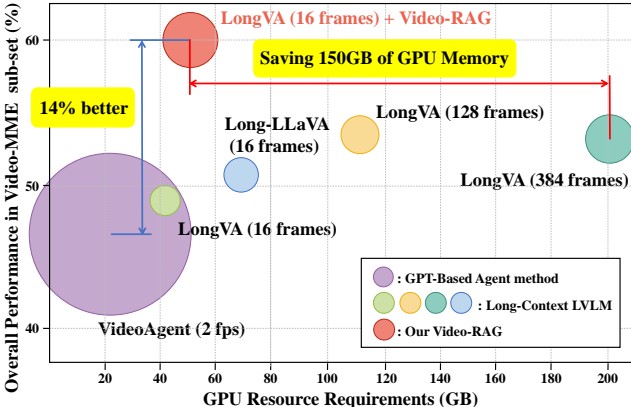

Figure 6: The comparison of our Video-RAG with two common approaches. The size of the bubbles represents the total time consumed for completing inference on the Video-MME [6] sub-set.

## D   Details of Similarity Score Calculation

In the process of using the RAG system to retrieve auxiliary texts extracted from videos, we define a similarity threshold $t$ to ensure the selection of relevant texts. Specifically, we employ FAISS-based [13] similarity to select OCR and ASR texts, while CLIP [30] similarity is used for keyframe selection. In our implementation, the similarity threshold $t$ is set to 0.3. As for OCR and ASR selection, For any given list of the retrieve request $\mathbf{R}$ and auxiliary texts $\mathbf{A}$, the Contriever [9] framework maps the text to a text embedding as:

$$\mathbf{E}_{a_i} = \texttt{Contriever}(\mathbf{A}_i), \quad i = 1, 2, \ldots, n$$
$$\mathbf{E}_{r_i} = \texttt{Contriever}(\mathbf{R}_i), \quad i = 1, 2, \ldots, n$$

The average embedding of the retrieve request is then computed as:

$$\mathbf{E}_r = \frac{1}{n} \sum_{i=1}^{n} \mathbf{E}_{r_i}$$

After that, the embedding of the request and the list of auxiliary texts is normalized:

$$\mathbf{E}_{a_i} = \frac{\mathbf{E}_{a_i}}{\| \mathbf{E}_{a_i} \|}, \quad \mathbf{E}_r = \frac{\mathbf{E}_r}{\| \mathbf{E}_r \|}$$

The similarity between the query embedding $\mathbf{E}_r$ and the document vector $\mathbf{E}_a$ is computed using the inner product, the FAISS library is employed to efficiently perform this search and return the indices of the auxiliary texts meeting the criterion:

$$S(\mathbf{E}_r, \mathbf{E}_{a_i}) = \mathbf{E}_r \cdot \mathbf{E}_{a_i} > t$$

As for object detection, we use CLIP to select the video keyframe. During this process, we first filter the object detection request $\mathbf{R}_{det}$ to ensure they correspond to CLIP-sensitive physical entities, avoiding the inclusion of abstract concepts. Specifically, if it is a single word, direct part-of-speech

filtering is applied; if it is a compound word, certain rules are followed to check for compliance, such as whether it is an adjective plus a noun, or a noun plus a noun. We use the Spacy library to achieve this. After this, we put the text "A picture of" before each object detection request.

Then, we extracting embedding from both the video frames $\mathbf{F}$ and the detection request $\mathbf{R}_{det}$:

$$\mathbf{E}_{\mathbf{F}_j} = \text{CLIP}(\mathbf{F}_j), \quad j = 1, 2, \ldots, m$$
$$\mathbf{E}_{\mathbf{R}_i} = \text{CLIP}(\mathbf{R}_{det_i}), \quad i = 1, 2, \ldots, n$$

The similarity between each video frame and the detection retrieve requests is computed using the dot product between the image and text feature embeddings. For each frame $\mathbf{F}_j$, and for each retrieve request $\mathbf{E}_{\mathbf{R}_i}$, the similarity score is given by:

$$S_{ij} = \mathbf{E}_{\mathbf{F}_j} \cdot \mathbf{E}_{\mathbf{R}_i}$$

where $\cdot$ denotes the dot product. The final similarity score for each frame is the average similarity across all requests:

$$S_j = \frac{1}{n} \sum_{i=1}^{n} S_{ij}$$

This computes the mean similarity for each frame across all text descriptions, resulting in a similarity vector $\mathbf{S} = [S_1, S_2, \ldots, S_m]$. The similarity scores are adjusted by a scaling factor $\alpha$, which is computed based on the number of frames $m$ and a base frame number $b$ (which is set to 16 and 4.0, respectively) to adapted different video sampling rate of LVLMs:

$$\alpha = \beta \times \frac{m}{b}$$

where $\beta$ is a predefined scaling parameter.

Next, the similarity scores are scaled and normalized to ensure that they sum to 1:

$$S_j^{\text{norm}} = \frac{\alpha \times S_j}{\sum_{k=1}^{m} S_k}$$

where $S_j^{\text{norm}}$ represents the normalized similarity score for frame $\mathbf{F}_j$.

The final step is to select the keyframes based on the normalized similarity scores. A threshold $t$ is applied to the normalized similarities, such that frames with similarity scores above the threshold are selected as keyframes:

$$\text{Keyframe:} \quad \mathbf{F}_j \quad \text{if} \quad S_j^{\text{norm}} > t$$

Thus, the set of selected keyframes is given by:

$$\mathbf{F}_{key} = \{\mathbf{F}_j \mid S_j^{\text{norm}} > t, \, j = 1, 2, \ldots, m\}$$

## E   More Ablation Studies

**Effect of different components of Video-RAG.** We evaluate the performance across sub-tasks within Video-MME [6], as shown in Figure 7. The results reveal that object detection auxiliary texts significantly enhance spatial perception and object counting, while OCR auxiliary texts specifically improve performance on text recognition tasks. Additionally, ASR auxiliary texts contribute to a general improvement in inference tasks, underscoring the critical role of audio transcription in video understanding. Given that audio transcription is considerably more time-consuming than character recognition or object detection, these texts should be selected based on the requirements of the application.

Besides studying the inference of different components of Video-RAG in the Video-MME [6] benchmark, we also experiment with a different type of video benchmark. We first evaluate LLaVA-Video in MLVU [54] and LongVideoBench [43] in both 7B and 72B scale with the 64-frame setting, results are shown in Table 9. As demonstrated, when all components are combined, we get optimal

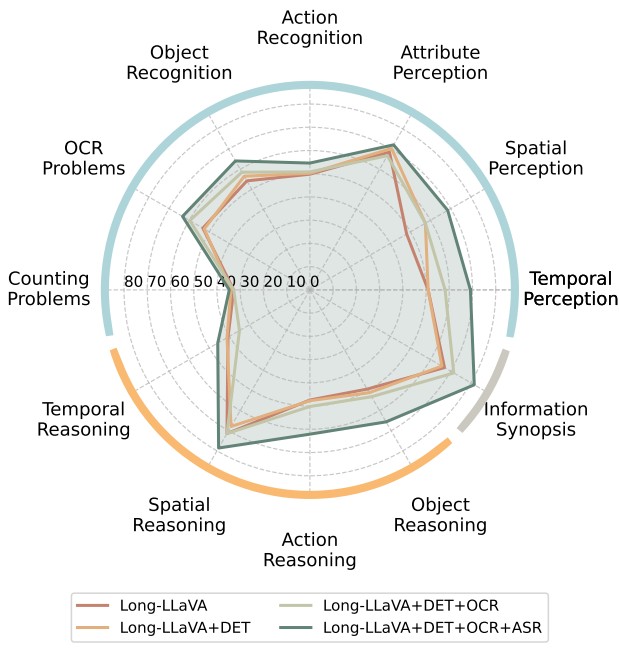

Figure 7: Performance on 12 sub-tasks in Video-MME [6] benchmark after applying different components in Long-LLaVA.

Table 9: Ablation study in MLVU and LongVideoBench.

| RAG | DET | OCR | ASR | 7B | | 72B | |
| | | | | MLVU | LVB | MLVU | LVB |
|---|---|---|---|---|---|---|---|
| | | | | 70.8 | 56.6 | 73.1 | 61.9 |
| ✓ | ✓ | | | 71.0 | 56.5 | 73.4 | 63.2 |
| ✓ | ✓ | ✓ | | 71.3 | 56.8 | 73.5 | 63.4 |
| ✓ | ✓ | ✓ | ✓ | **72.4** | **58.7** | **73.8** | **65.4** |
| | ✓ | ✓ | ✓ | 70.3 | 58.3 | 72.9 | 64.0 |

performance in both datasets, including 7B and 72B scales. Specifically, the performance in MLVU [54] even declined when the RAG system was not implemented.

Then, to better point out the role of DET and OCR, we evaluate Video-RAG in VNBench [53] with Long-LLaVA-7B [45]. VNBench is a synthetic benchmark designed to evaluate models' long-context abilities, covering tasks such as retrieval, ordering, and counting. VNBench randomly inserts stickers or text into the video that has nothing to do with the original content of the video, thus typically challenging the model's needle-in-the-haystack capability. As shown in Table 10, we find that applying DET and OCR as auxiliary texts can significantly improve the performance in retrieval, ordering, and counting tasks. However, the ASR component will decline the performance due to the subtitles are not ancillary to this particular task. These results demonstrated that our proposed distinct types of auxiliary texts can be selected according to the application needs to meet the requirements better.

## F   More Qualitative Results

In this section, we show more results of LLaVA-Vdieo-7B when applying Video-RAG in different examples in Figure 9. The figure highlights several representative cases involving detailed video comprehension from Video-MME [6]. As illustrated, augmenting LLaVA-Video with external tools to process and retrieve auxiliary texts from videos significantly enhances its ability to reduce visual hallucinations, thereby enabling more accurate and confident responses to user queries.

## Decouple Prompt of the Multiple-choice Question

```
To answer the question step by step, list all the physical entities related to
the question you want to retrieve, you can provide your retrieve request to
assist you by the following JSON format:
{
     "ASR": Optional[str]. The subtitles of the video that may relavent to the
question you want to retrieve, in two sentences. If you no need for this
information, please return null.
     "DET": Optional[list]. (The output must include only physical entities, not
abstract concepts, less than five entities) All the physical entities and their
location related to the question you want to retrieve, not abstract concepts. If
you no need for this information, please return null.
     "TYPE": Optional[list]. (The output must be specified as null or a list
containing only one or more of the following strings: 'location', 'number',
'relation'. No other values are valid for this field) The information you want
to obtain about the detected objects. If you need the object location in the
video frame, output "location"; if you need the number of specific object,
output "number"; if you need the positional relationship between objects, output
"relation".
}
## Example 1:
Question: How many blue balloons are over the long table in the middle of the
room at the end of this video? A. 1. B. 2. C. 3. D. 4.

Your retrieve can be:

{
     "ASR": "The location and the color of balloons, the number of the blue
balloons.",
     "DET": ["blue ballons", "long table"],
     "TYPE": ["relation", "number"]
}
## Example 2:
Question: In the lower left corner of the video, what color is the woman wearing
on the right side of the man in black clothes? A. Blue. B. White. C. Red. D.
Yellow.

Your retrieve can be:

{
     "ASR": null,
     "DET": ["the man in black", "woman"],
     "TYPE": ["location", "relation"]
}
## Example 3:
Question: In which country is the comedy featured in the video recognized
worldwide? A. China. B. UK. C. Germany. D. United States.

Your retrieve can be:

{
     "ASR": "The country recognized worldwide for its comedy.",
     "DET": null,
     "TYPE": null
}
Note that you don't need to answer the question in this step, so you don't need
any infomation about the video of image. You only need to provide your retrieve
request (it's optional), and I will help you retrieve the infomation you want.
Please provide the json format.
```

Figure 8: Decouple prompt of the multiple-choice question for LVLMs.

Table 10: Results on combinations of different auxiliary texts in VNBench [53] with 1-try setting when applying 7B Long-LLaVA [45] as LVLM under the 32-frames setting. **Ret**, **Ord**, and **Cnt** represent retrieval, ordering, and counting tasks, respectively.

| RAG | DET | OCR | ASR | Ret | Ord | Cnt | Overall |
|:---:|:---:|:---:|:---:|:---:|:---:|:---:|:---:|
| | | | | 65.1 | 25.6 | 24.2 | 38.3 |
| ✓ | ✓ | | | 66.9 | 28.4 | 23.8 | 39.7 |
| ✓ | ✓ | ✓ | | 68.2 | 31.3 | 28.9 | 42.8 |
| ✓ | ✓ | ✓ | ✓ | 66.7 | 31.3 | 29.6 | 42.5 |

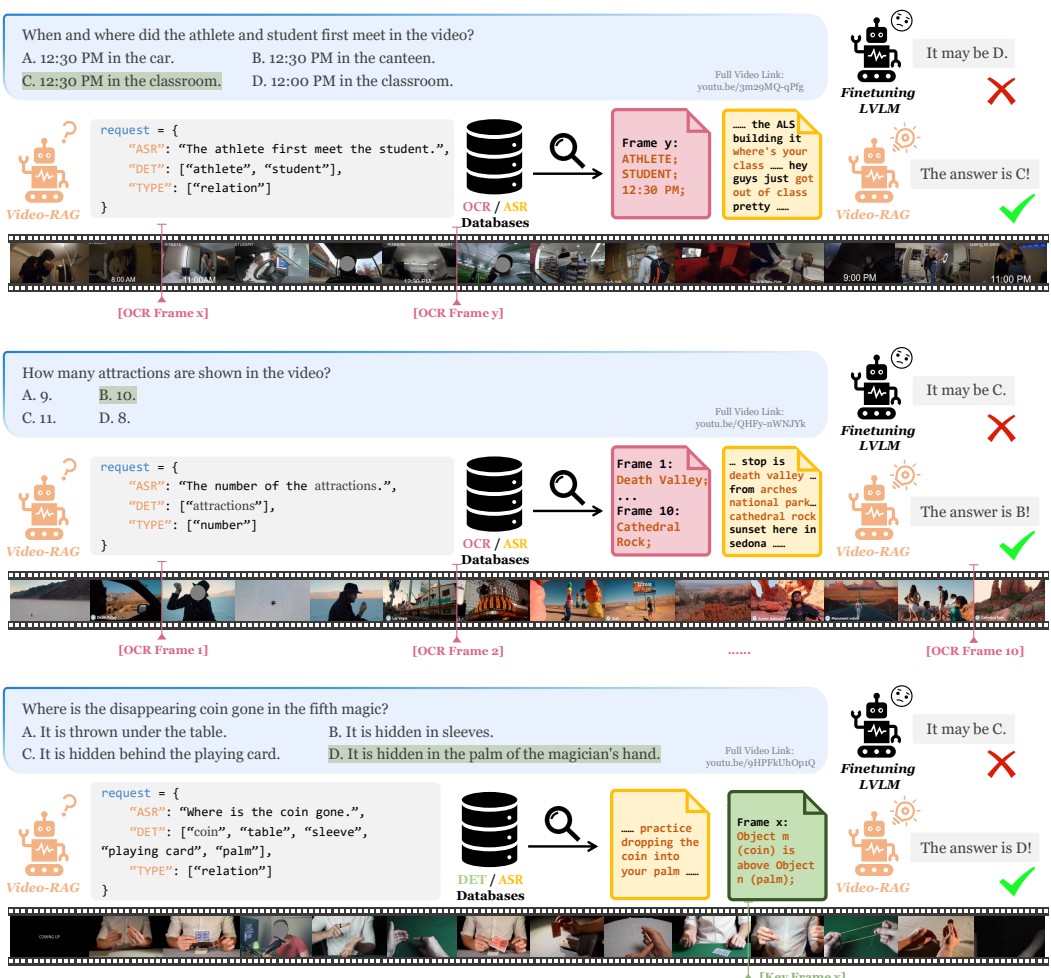

Figure 9: Qualitative results of LLaVA-Vdieo when applying Video-RAG.

