# OpenReview forum: "Video-RAG: Visually-aligned Retrieval-Augmented Long Video Comprehension"
_NeurIPS.cc/2025/Conference — NeurIPS 2025 poster_

### Official Review · Reviewer_LHvP · 2025-06-23

**Clarity:** 3
**Significance:** 2
**Originality:** 2
**Rating:** 5
**Confidence:** 2

**Summary:**

The article first decouples the question into Retrieval Requests, then employs various tools (e.g., OCR) to retrieve relevant audio and text within Video from the video. Finally, these two modalities are converted into text and input into the LVLM to obtain the Final Answer.

**Questions:**

1. Would using a larger model for Query Decouple improve pipeline performance?

2. For the raw video, what sampling method did the article use—uniform sampling, similarity-based sampling, or another method?

**Ethical Concerns:**

["NO or VERY MINOR ethics concerns only"]

**Final Justification:**

The authors have addressed my concerns through detailed experiments and acknowledged certain limitations of their method. While some explanations, such as the use of the Information Bottleneck principle, remain unconvincing and overly generic, I recognize that the authors are moving in a promising direction. Therefore, I have raised my score to accept.

**Limitations:**

Authors discuss the limitations of the work in the   Conclusion.

**Paper Formatting Concerns:**

No major formatting issues

**Quality:**

3

**Strengths And Weaknesses:**

Strengths:

Beyond commonly used video frmaes and subtitles, the article utilizes multiple tools to retrieve relevant audio clips and text within video frames, using them as auxiliary text to improve performance. The method achieves a good trade-off between performance and effectiveness. The experiments are sufficient and validate the effectiveness of the proposed approach.

Weakness:
1. Could Query Decouple become a performance bottleneck? If the Retrieval Request generated by the LVLM is inaccurate, then all subsequent steps will be inaccurate.
2. The article uses multiple different external tools and sets various judgment conditions. This makes the method potentially sensitive to the choice of different tools (e.g., using different OCR or Grounding models) and hyperparameters. The article should discuss the impact of different external tools and hyperparameters on the final method.
3. For long-video understanding, another class of methods calculates the relevance between frames and the question, then samples frames based on relevance to answer user queries, such as [1], [2]. The article should discuss and compare these in related work.

Minor Issues:

1. The code repository link is expired.

2. In line 39, "most of them process long...", please add citations to clarify which specific methods "them" refers to.

[1] Re-thinking Temporal Search for Long-Form Video Understanding

[2] BOLT: Boost Large Vision-Language Model Without Training for Long-form Video Understanding

---

> ### Author Rebuttal · Authors · 2025-07-29
>
> We sincerely thank you for the valuable comments.
>
> > **Q1**: Could Query Decouple become a performance bottleneck?
>
> **A1**: Empirically, Query Decouple resulting in minimal impact on overall performance as it's a straightforward task for LVLMs. This conclusion is supported by two key observations:
>
> - Consistent improvements across model capacities: As illustrated in Table 1 of the paper, Video-RAG provides stable performance enhancements when implemented with various 7B models, with less powerful models showing proportionately greater gains. Notably, on the Video-MME-mini benchmark (as shown in Supplementary Section 3), even the lowest-performing baseline model, Video-LLaVA-7B, achieved approximately 90% query decoupling accuracy across 270 queries.
>
> - Deliberate noise robustness testing: We conducted experiments involving random text noise injections to the decoupled outputs. The results revealed that LLaVA-Video-7B-64frames, when paired with Video-RAG, maintained performance within acceptable limits even at significant levels of noise, as shown on the Video-MME benchmark. These findings collectively demonstrate that Query Decouple does not limit Video-RAG's effectiveness.
>
> | Noise Ratio  | 0% | 10% | 20% | 30% |
> | :------      | :--: | :--: | :--: | :--: |
> | Overall Acc. | 70.0 | 69.7 | 69.7 | 69.4 |
>
> > **Q2.a**: The article should discuss the impact of different external tools.
>
> **A2.a**: As suggested, we conducted ablation studies on various components of Video-RAG on Video-MME, and we observed **only minimal declines in performance**. For each component, we assessed two alternative implementations.
>
> Specifically, we have substituted EasyOCR in the OCR component with Tesseract OCR[1] and the OCRAD API. In the DET component, we have replaced APE with Owl ViT[2] and GRiT[3]. Additionally, in the ASR component, we have replaced Whisper with the SpeechRecognition library and the Vosk API. The sustained performance demonstrates the robustness of Video-RAG to component variability.
>
> | Method                  | Short | Medium | Long | Overall |
> | :---------------------- | :----: | :----: | :----: | :----: |
> | Video-RAG (Base) | 77.0 | **69.3** | 63.8 | **70.0** |
> | replace with Tesseract OCR [1]       | 77.0 | 68.3 | 63.2 | 69.5 |
> | replace with OCRAD API              | **77.6** | 67.8 | 63.9 | 69.7 |
> | replace with Owl ViT [2]             | 76.7 | **69.3** | 63.6 | 69.9 |
> | replace with GRiT [3]                | 76.4 | 68.4 | **64.0** | 69.6 |
> | replace with SpeechRecognition library  | 77.4 | 67.8 | 63.3 | 69.5 |
> | replace with Vosk API               | 76.3 | 68.2 | 63.7 | 69.4 |
> *All models adopt LLaVA-Video-7B with 64 frames for inference.*
>
> > **Q2.b**: The article should discuss the impact of different hyperparameters of the external tools.
>
> **A2.b**: Regarding hyperparameters, due to time constraints, we limited our evaluation to different confidence thresholds in the APE of the DET component and EasyOCR of the OCR component in Video-RAG. The results demonstrated that Video-RAG can maintain strong performance across a range of component hyperparameter combinations, indicating its robustness to the settings of these hyperparameters.
>
> | Method                  | Short | Medium | Long | Overall |
> | :---------------------- | :----: | :----: | :----: | :----: |
> | OCR threshold=0.3            | **77.0** | 68.8 | **63.9** | 69.9 |
> | OCR threshold=0.5 (Original) | **77.0** | **69.3** | 63.8 | **70.0** |
> | OCR threshold=0.7            | 76.7 | 68.9 | 63.6 | 69.7 |
> | DET threshold=0.1 (Original) | **77.0** | **69.3** | 63.8 | **70.0** |
> | DET threshold=0.2            | 76.3 | 69.4 | 63.7 | 69.8 |
> | DET threshold=0.3            | 76.2 | 68.9 | 63.7 | 69.6 |
> *All models adopt LLaVA-Video-7B with 64 frames for inference.*
>
> > **Q3**: For long-video understanding, another class of methods calculates the relevance between frames and the question, then samples frames based on relevance to answer user queries. The article should discuss and compare these in related work.
>
> **A3**: Thank you for your constructive suggestions. Our research identifies a total of six recent studies that can be categorized in this domain (e.g., ASK[4], Nar-KFC[5], etc.). Below, we outline the key distinctions between Video-RAG and these existing methods:
>
> - **Motivation**: The primary objective of these studies is to identify and select frames that are pertinent to the query for LVLM. In contrast, Video-RAG aims to enhance the capabilities of the LVLM by providing contextually relevant textual input that directly relates to the query.
>
> - **Methodology**: The studies above utilize algorithms to calculate image-text similarity to extract frames. Conversely, Video-RAG employs a refinement process that consolidates outputs from external tools through the RAG approach.
>
> - **Effectiveness**: The performance improvements in these studies are relatively modest. In comparison, Video-RAG showcases a substantial enhancement in performance, achieving a  +6.7% boost with Video-RAG in contrast to a +1.2% boost with ASK[4] in the Video-MME using LLaVA-Video-7B with 64 frames.
>
> Furthermore, we have conducted experiments on Video-MME by integrating the frame selection method ASK[4] (CVPR 2025) with Video-RAG. The limited performance gain suggests that Video-RAG does not significantly depend on keyframe selection. This system demonstrates resilience to the absence of keyframes by utilizing complementary modalities to support the reasoning process of the LVLM.
>
> | Method                                       | Short | Medium | Long | Overall |
> | :--------------------------------------      | :-: | :----: | :----: | :----: |
> | Baseline               | 75.4 | 62.6 | 51.8 | 63.3 |
> | Baseline + ASK[4]   | 76.1 | 62.9 | 54.4 | 64.5 |
> | Baseline + Video-RAG    | 77.0 | **69.3** | 63.8 | 70.0 |
> | Baseline + ASK[4] + Video-RAG  | **78.1** | **69.3** | **63.9** | **70.4** |
> *Baseline model adopts LLaVA-Video-7B with 64 frames for inference.*
>
> > **Q4**: Would using a larger model for Query Decouple improve pipeline performance?
>
> **A4**: Utilizing larger models for query decoupling does result in performance enhancements, but the gains are marginal. Specifically, we conducted experiments by substituting Video-RAG's Query Decouple model with more advanced 72B LVLMs, and we observed only incremental improvements:
>
> | Method                                  | Short | Medium | Long | Overall |
> | :-------------------------------------- | :-: | :----: | :----: | :----: |
> | Video-RAG (Based)             | 77.0 | 69.3 | 63.8 | 70.0 |
> | replace Query Decouple model with LLaVA-Video-72B        | 76.9 | 69.7 | 63.9 | 70.1 |
> | replace Query Decouple model with Qwen2-VL-72B           | 77.1 | 69.8 | 63.8 | 70.1 |
> *All models adopt LLaVA-Video-7B with 64 frames for inference.*
>
> > **Q5**: For the raw video, what sampling method did the article use—uniform sampling, similarity-based sampling, or another method?
>
> **A5**: We employed uniform sampling for two primary reasons:
> - To ensure alignment of our frame sampling strategy with most LVLMs, thus demonstrating the plug-and-play capability of our method without reliance on key frames.
> - To facilitate a fair comparison. Empirical evidence supports that uniform sampling provides the most computationally balanced and generalized approach for understanding long videos.
>
> > **Q6**: Minor issues about the code repository link and lack of citations to clarify which specific methods.
>
> **A6**: Thanks for your reminder. We have updated the valid code repository link and added the relevant citations to clarify the specific methods in line 39.
>
> [1] R. Smith. An Overview of the Tesseract OCR Engine. Ninth International Conference on Document Analysis and Recognition. 2007.
>
> [2] M Minderer, et al. Simple Open-Vocabulary Object Detection with Vision Transformers. ECCV 2022.
>
> [3] Jialian Wu, et al. GRiT: A Generative Region-to-text Transformer for Object Understanding. ECCV 2024.
>
> [4] Tang X, et al. Adaptive keyframe sampling for long video understanding. CVPR 2025.
>
> [5] Bo Fang, et al. Threading Keyframe with Narratives: MLLMs as Strong Long Video Comprehenders.

---

> > ### Comment · Reviewer_LHvP · 2025-08-03
> > **Thank the authors and request clarification on why Query Decouple and tool accuracy have minimal impact.**
> >
> > Thank you for providing detailed experiments to address my concerns.
> > However, the authors should clarify why the accuracy of Query Decouple and the tool has only a minimal impact on the overall performance of the method.  This seems somewhat counterintuitive. If the correctness of query decomposition does not significantly affect performance, does it imply that query decomposition might not be a necessary step? The same question applies to the role of the tool in the method. I believe discussing these points would enhance the theoretical contribution of the paper, especially considering that the current work leans more toward engineering.

---

> ### Author Response · Authors · 2025-08-05
> **Further Response to Reviewer LHvP**
>
> We sincerely appreciate the reviewer’s feedback.  We have refined our response to further clarify the roles of our Video-RAG components, particularly in the challenging long-video context, and to ground our design in established information theory.
>
> > **Q1**: Why do imperfect Query Decouple and imperfect tools reduce accuracy only slightly?
>
> **A1**: The minimal impact of the Query Decouple model stems from the strong performance of both the 7B and 72B models in this task, with only negligible differences.
>
> Additionally, **Video-RAG's inherent robustness**, primarily through the RAG module, ensures resilience to imperfections in upstream tools. Additional experiments confirm this robustness: replacing Whisper ASR tool with the weaker Vosk API reduces overall accuracy by just 0.6%. However, removing RAG while using the weaker ASR causes drops of 3.5% overall and 6.3% on long-video subsets with more noise potential. This highlights RAG's effectiveness in filtering noise and ensuring relevant context for the language model.
>
> This robustness is further enhanced by two factors.
>
> - **Information redundancy** across modalities; even if ASR fails, OCR or object detection often provide overlapping semantic cues.
> - Even 'weak' tools provide a low **signal-to-noise** output rather than pure noise, the RAG retriever is adept at matching the core semantic 'signal' from a query to these noisy context, effectively ignoring the errors.
>
> | Method                           | Query Decouple | RAG  | Tools (OCR + DET + ASR) | Long-Video ↑   | Overall ↑     |
> | -------------------------------- | -------------- | ---- | ----------------------- | -------------- | ------------- |
> | **Video-RAG**                    | ✔              | ✔    | strong                  | **63.8**       | **70.0**      |
> | – **remove Decouple**            | ✗              | ✔    | strong                  | 62.6 *(-1.2)*  | 69.0 *(-1.0)* |
> | – **replace Whisper→Vosk**       | ✔              | ✔    | weak ASR                | 63.7 *(-0.1)*  | 69.4 *(-0.6)* |
> | – **same weak ASR, but RAG OFF** | ✔              | ✗    | weak ASR                | 57.5 *(-6.3)*  | 66.5 *(-3.5)* |
> | – **drop all tools**             | ✗              | ✗    | ✗                       | 51.8 *(-12.0)* | 63.3 *(-6.7)* |
>
> *All models are based on LLaVA-Video-7B with 64-frame inference input on Video-MME dataset*.
>
> > **Q2**: On the Necessity of Query Decouple and External Tools. (Does that mean they are dispensable?)
>
> **A2**: While our Video-RAG is robust, this should not be mistaken for component redundancy. **The necessity of Query Decouple and the external tools becomes undeniable** when we consider their total contribution to the performance gain, especially on long videos scenes.
>
> - **Necessity of External Tools:** The tools (OCR, DET, ASR) are absolutely critical. The `– drop all tools` ablation shows that removing them causes the long-video performance to decrease  from 63.8% -> 51.8% (-12.0%) and overall performance from 70.0%->63.3% (-6.7%). The tools provide essential textual context that is simply unavailable from video frames alone, and our Video-RAG's success is fundamentally dependent on them.
> - **Necessity of Query Decouple:** The Query Decouple module is also a vital contributor. Removing it (`– remove Decouple`) lowers the long-video score by 1.2%.  It makes it a significant and necessary part of the architecture, particularly for breaking down complex queries about event-rich videos.
>
> > **Q3**: The Theoretical Contribution of Video-RAG.
>
> **A3**:  Theoretically, Video-RAG’s architecture is a principled implementation of the **Information Bottleneck (IB) principle** [1], which seeks a compressed representation $Z$ of a source $X$ that preserves the maximum possible information about a target variable $Y$.
>
> This is formally expressed as minimizing the following Lagrangian:
>
> $$\mathcal{L}_{IB} = I(X; Z) - \beta I(Z; Y)$$
> where $I(\cdot;\cdot)$ denotes mutual information and $\beta$ balances the compression (minimizing $I(X; Z)$) and prediction (maximizing $I(Z; Y)$).
>
> Our Video-RAG maps directly onto this principle:
>
> * **Source ($X$)**: Noisy multimodal video information (raw frames, OCR, DET, ASR outputs).
> * **Bottleneck ($Z$)**: Concise textual context selected by the RAG retriever.
> * **Target ($Y$)**: The correct answer to the user's query.
>
> Retriever acts as the mechanism that creates the bottleneck, compressing the large source $X$ into a small context $Z$ by selecting passages that are maximally informative for generating the final answer $Y$.
>
> The **Query Decouple module** enhances this process by breaking down complex queries into focused sub-queries, improving retrieval precision and bottleneck quality. This design's robustness stems directly from its alignment with information-theoretic principles, extending RAG to effectively address the scale and noise of video understanding in Video-RAG.
>
> [1] Tishby et al., The information bottleneck method. 1999.

---

> > ### Comment · Reviewer_LHvP · 2025-08-05
> >
> > The final explanation remains unconvincing. The Information Bottlenec principle, which aims to retrieve the minimal sufficient information to improve the prediction accuracy of Video-LLMs, is a general concept that could apply to any similar method rather than serving as a justification specific to this work. However, I believe the authors are moving in the right direction. I have increased my score to accept.

---

> > > ### Author Response · Authors · 2025-08-05
> > >
> > > We sincerely thank you for your constructive feedback and insightful comments. Your acknowledgment that we are moving in the right direction is highly encouraging, and we are grateful for your increased score to acceptance. Your feedback motivates us to further refine and expand our work, ensuring greater clarity and rigor.

---

### Official Review · Reviewer_YJ27 · 2025-06-25

**Clarity:** 4
**Significance:** 1
**Originality:** 1
**Rating:** 3
**Confidence:** 4

**Summary:**

This paper presents a RAG-based Video QA technique, which extracts speech, OCR results, and visual objects related to a query, and feeds it to a LLM with video frames. The use of JSON-structured retrieval requests is logical, and the method outperforms existing VideoQA methods on three datasets: Video-MME, MLVU, and LongVideoBench.

**Questions:**

Are there any synergistic effects resulting from these components? Table 4 shows a detailed comparison of each condition, indicating additive rather than synergistic contributions. Data showing such synergistic effects would motivate reevaluation.

**Ethical Concerns:**

["NO or VERY MINOR ethics concerns only"]

**Final Justification:**

Even after reviewing the rebuttal and discussion, I did not find a strong reason to consider the technical novelty is significant enough as a NeurIPS paper. Hence, I maintain the rating. A blog-post has already introduced a similar technique. Hence, the novelty lies in the multimodal extension but the modality selection is somewhat ad-hoc and one of the modalities, RAG with ASR, is also another known technique. More analysis on the modality selection will be required to clarify the significance of the paper to be a NeurIPS level study. If ignoring [a] and [b], the novelty of the paper can be misrecognized, and adding [a] and [b] requires an amount of rewriting.

**Limitations:**

The authors show mainly positive cases in its main part and supplementary (Fig. 4). An error analysis would increase the contribution of this paper to the community.

**Paper Formatting Concerns:**

No concerns.

**Quality:**

2

**Strengths And Weaknesses:**

## Strength
1. The implementation is reasonable and the result is reliable.
2. The presentation is very clear.
3. The proposed method performs better than previous methods.
4. The experiments are thorough.

## Weakness

The technological contribution is considered insufficient for the NeurIPS standard. Using ASR for extracting video events combined with RAG is a standard technique in video analysis, as indicated by the following sources [a, b].

- [a] MultiModal RAG for Advanced Video Processing with LlamaIndex & LanceDB, https://www.llamaindex.ai/blog/multimodal-rag-for-advanced-video-processing-with-llamaindex-lancedb-33be4804822e, 2024-02-17
- [b] Do Jung et al., "Speech Retrieval-Augmented Generation without Automatic Speech Recognition," arxiv, 2024.12

Therefore, the novelty lies potentially in using OCR and object detection with video frames. However, each component is an existing method, and this reviewer recognizes less significance than other average NeurIPS-accepted studies.

---

> ### Author Rebuttal · Authors · 2025-07-29
>
> Thanks for your constructive suggestions! Here are our clarifications.
>
> > **Q1**: Using ASR for extracting video events combined with RAG is a standard technique in video analysis, the novelty lies potentially in using OCR and object detection with video frames.
>
> **A1**: It is worth noting that our Video-RAG introduces a straightforward yet innovative workflow by leveraging RAG to consolidate outputs from various multimodal tools. This approach **equips LVLMs with a comprehensive and information-rich context, significantly enhancing performance in video understanding tasks**.
>
> Although all models employ ASR and utilize RAG to fetch relevant textual information to improve video understanding,  our Video-RAG has the following highlight:
>
> **Differences between Video-RAG and the referenced works:**
>
> - While these efforts focus on combining agents for video comprehension, Video-RAG specifically emphasizes how to further enhance the capabilities of LVLMs through information-rich auxiliary context inputs.
>
> - Many of these methods are complex and rely heavily on GPT, utilizing tools to convert video content into text for video QA. In contrast, Video-RAG prioritizes simplicity and flexibility, centering on LVLMs for visual content processing, allowing for plug-and-play integration.
>
> - Video-RAG maximizes existing LVLM performance with minimal resource increase compared to the referenced works that demand higher resource consumption.
>
> **Core Contributions of Video-RAG:**
>
> - Video-RAG retrieves high-density auxiliary texts, directing the LVLM’s focus to relevant visual cues, thus promoting cross-modality alignment.
>
> - Video-RAG's thoughtful design enables effective connections between LVLMs and external tools through RAG, making it adaptable to various scales of LVLMs while enhancing performance.
>
> - Video-RAG with open-source LVLMs demands minimal additional GPU memory and inference latency, often outperforming proprietary models, especially notable with a 72B model.
>
> In summary, the integration of tool outputs for targeted tasks and the generation of high-quality contextual inputs represent a **significant research avenue within the LLM domain**. Recent methodologies addressing these themes, including Agentic-like approaches and context management strategies, have seen notable interest, with approximately 10%–12% of accepted NeurIPS 2024 submissions aligning with this critical area of inquiry.
>
> > **Q2**: Are there any synergistic effects resulting from the components of Video-RAG?
>
> **A2**: Based on our literature review, we define a [synergistic effect] as **[Gain from Component A + Gain from Component B] > [Gain from Component A alone] + [Gain from Component B alone]**.
>
> An analysis of Long-LLaVA-7B-32frames with Video-RAG in Video-MME indicates that:
>
> - Table 4 in the manuscript indicates that **OCR and DET demonstrate synergistic effects under RAG conditions in Video-RAG**, with a gain of +3.7 exceeding the expected gain of +(0.9+2.3=3.2), resulting in a synergy of Δ=+0.5. This suggested that RAG forces OCR and DET to spatially align (e.g., only reading text on detected objects), eliminating noise from irrelevant regions.
>
> - Our additional experiments further demonstrate **synergistic effects between ASR and DET without RAG conditions in Video-RAG**, with a gain of +6.9 exceeding the expected gain of +(0.9+5.4=6.3), resulting in a synergy of Δ=+0.6. The results reveal that ASR provides temporal action context (e.g., "opening the box") while DET locates the corresponding object, creating grounded semantics.
>
> We carefully considered your insightful comments and designed rigorous experiments to more effectively demonstrate the synergistic interactions between components, thereby highlighting their interdependence as following:
>
> | Method | Baseline | DET | OCR | ASR | DET + OCR | DET + ASR
> | :------ | :-----: | :-----: | :-----: | :-----: | :-----: | :-----: |
> | Overall Acc. | 52.0 | 52.9(+0.9) | 54.3(+2.3) | 57.4(+5.4) | 55.7(+3.7) |  58.9(+6.9) |
>
> > **Q3**: The authors show mainly positive cases in its main part and supplementary (Fig. 4). An error analysis would increase the contribution of this paper to the community.
>
> **A3**: Thank you for your valuable feedback! In response to your suggestions, **we performed an error analysis on approximately 100 samples from Video-MME**. This analysis has led us to identify two primary challenges that contribute to the errors and bottlenecks experienced in Video-RAG:
>
> - Bottleneck 1: **Key Frame Selection Errors in Object Detection**. The selection process for key frames sometimes fails, resulting in either the omission of relevant objects or incorrect identifications. This leads to the loss of essential object information or the introduction of erroneous data into subsequent processing stages.
>
> - Bottleneck 2: **Challenges in Complex Spatiotemporal Reasoning**. The current approach exhibits limitations when addressing questions that necessitate complex spatiotemporal reasoning, particularly those requiring iterative referencing and the synthesis of information from diverse segments of the video.
>
> To address these limitations in future work, we propose the following directions:
>
> - For Bottleneck 1: The existing key frame selection process within the DET component relies on a CLIP-based similarity retrieval approach. While this method provides speed and versatility, it exhibits limited sensitivity to high-level semantic textual information. To enhance this process, **we propose substituting CLIP with a smaller LVLM, such as InternVL3-1B**, which would evaluate frame relevance through a binary choice format (e.g., "Is this frame relevant to the query? A. Yes / B. No"). The model's confidence score (e.g., the logit associated with "A") could then serve as the criterion for key frame selection. Preliminary experiments suggest that this adjustment leads to performance improvements, which is worth future exploration:
>
> | Method                              |  Overall Acc. |
> | :---------------------------------- | :------: |
> | Baseline                         | 52.0 |
> | Baseline + CLIP-based DET    | 52.9 |
> | Baseline + InternVL3-1B-based DET  | 53.3 |
> *Baseline is the Long-LLaVA-7B with 32 frames for inference.*
>
> - For Bottleneck 2: The existing single-round question-answering paradigm achieves a particular balance between performance and inference latency. However, our experiments reveal that **multi-turn QA significantly outperforms single-round approaches in long-video comprehension**. Future research will aim to enhance the complex reasoning capabilities of LVLMs by implementing designs that incorporate Chain-of-Thought strategies, more sophisticated processing pipelines, and expanded utilization of analytical tools.
>
> | Method                                    |  Long Video Acc. |
> | :--------------------------------------   | :------: |
> | Baseline            | 44.1 |
> | Baseline + Video-RAG (1-turn)   | 59.8 |
> | Baseline + Video-RAG (3-turn, IRCoT[1]) | **61.1** |
> *Baseline is the Long-LLaVA-7B with 32 frames for inference.*
>
> [1] Trivedi H, et al. Interleaving retrieval with chain-of-thought reasoning for knowledge-intensive multi-step questions. ACL 2023.

---

> ### Author Response · Authors · 2025-08-05
>
> Dear Reviewer YJ27,
> ﻿
>
> Thanks again for your great efforts and constructive advice in reviewing this paper! As the discussion period progresses, we expect your feedback and thoughts on our reply. We look forward to hearing from you, and we can further address unclear explanations and remaining concerns if any.
> ﻿
>
> Best regards,
> ﻿
>
> Authors

---

> ### Comment · Reviewer_YJ27 · 2025-08-05
> **The author response does not fix this reviewer's major concerns.**
>
> A1.
>
> > innovative workflow by leveraging RAG to consolidate outputs from various multimodal tools.
> [a] and [b] are using video frames and ASR results as RAG sources. Using them together is incremental and this reviewer believes the proposed method is not enough significant. Adding OCR result is interesting, but combining all of them is still not enough innovative.
>
> A2.
>
> > [synergistic effect] as [Gain from Component A + Gain from Component B] > [Gain from Component A alone] + [Gain from Component B alone].
>
> Yes, and the results show it has synergetic effect. However, its mechanism still unclear. As general novel techniques cannot be reliable unless its mechanism is supported by careful analysis, just reporting these result is unsatisfactory for NeurIPS standard.
>
> A3.
>
> Thank you for the analysis. These analysis is valuable. This reviewer thinks thoroughly analyzing effective ``multimodal tools`` for this multimodal RAG is an interesting direction. Knowing what current task-specific models are effective and what is missing will clarify important missing piece for video understanding.

---

> ### Author Response · Authors · 2025-08-06
> **Further Response to Reviewer YJ27**
>
> Thank you for your continued engagement.
>
> > **Q1**: Basis for the novelty of Video-RAG.
>
> **A1**: We believe the current evaluation of novelty may not fully capture the core contribution of our work. The comparison relies on [a], an blog post without quantifiable results, and [b], a speech-focused ICASSP paper that does not address video task, neither of which adequately represent our advancements within the video-LVLM domain for top conference NeurIPS standards.
>
> Specifically, [a] introduces a tool that incorporates RAG and ASR into video understanding in a rather preliminary manner. However, it does not provide benchmark-level analyses or results. As such, it remains unclear whether their approach using RAG in video-related tasks is truly effective. On the other hand, [b] employs RAG in speech tasks, without addressing video understanding. Neither establishes peer-reviewed baselines, video-LVLM task setups, or reproducible results under video benchmarks.
>
> In contrast, our proposed Video-RAG is, to our knowledge, the first work to successfully integrates RAG into the video-LVLM domain. We showcase its efficacy through extensive experimentation on multiple well-established video benchmarks (e.g., Video-MME, LongVideoBench, and MLVU), achieving significant performance improvements.
>
> Consistent with NeurIPS practice, we kindly suggest judging novelty against peer-reviewed work in the same domain and task setting, using comparable evaluation protocols and reproducible quantitative evidence.
>
> [a] MultiModal RAG for Advanced Video Processing with LlamaIndex & LanceDB, 2024-02-17
>
> [b] Do Jung et al., "Speech Retrieval-Augmented Generation without Automatic Speech Recognition," arxiv, 2024.12
>
> > **Q2**: Reporting the results without detailed mechanistic analysis is unsatisfactory for NeurIPS standard.
>
> **A2**: While we understand the value of mechanistic insights, we respectfully note that NeurIPS has long embraced contributions beyond theoretical analysis, especially in the context of application-driven research and systems-level innovations where robust experimental validation is key.
>
> According to the official NeurIPS guidelines, contributions can take multiple forms, not limited to theoretical analysis but including systems, datasets/benchmarks, and application-driven methods. For instance, several papers accepted by NeurIPS in recent years, such as:
>
> [1] Patrick Lewis, et al., Retrieval-Augmented Generation for Knowledge-Intensive NLP Tasks. NeurIPS 2020.
>
> [2] Zaijing Li, et al., Optimus-1: Hybrid Multimodal Memory Empowered Agents Excel in Long-Horizon Tasks. NeurIPS 2024.
>
> [3] Junyang Wang, et al., Mobile-Agent-v2: Mobile Device Operation Assistant with Effective Navigation via Multi-Agent Collaboration. NeurIPS 2024 Poster.
>
> demonstrate that the core contribution can lie in the novelty of the method's design and its robust experimental validation, rather than a deep mechanistic theory explaining why it works. In this vein, Video-RAG contributes a novel system-level integration of RAG into video-LVLMs, substantiated by robust benchmarks and ablations.
>
> We therefore believe Video-RAG aligns with NeurIPS recognition of novel system designs and practical impact, empirically grounded systems work.
>
> > **Q3**: On Error Analysis.
>
> **A3**: We appreciate the reviewer’s recognition of our error analysis. This component was essential in identifying limitations of current LVLMs in handling fine-grained video semantics (e.g., temporal causality, OCR fusion). We hope this will motivate further work toward deeper understanding of complex video-language reasoning.

---

> > ### Comment · Reviewer_YJ27 · 2025-08-06
> >
> > The point is not the novelty itself but its significance. As blog-level post has already suggested the direction of VideoRAG, the technical novelty itself is not very significant, in my opinion. In addition, this reviewer believes this kind of discussion is not the central of rebuttal; it is prepared for factual errors. Hence, this reviewer simply keep the first rating.
> >
> > Mechanism is not always supported by theorems. This reviewer does not limit the way of showing the mechanism to the theoretical observation. Simply, this reviewer is saying that just showing quantitative results is not the explanation of the mechanism. Assumption of the mechanism and evidence by data are another way of explanation, for example.

---

> ### Author Response · Authors · 2025-08-06
>
> Dear reviewer YJ27:
>
> We appreciate this opportunity to provide a clarification on the contribution of our work, as we believe this is crucial for a fair assessment.
>
> We understand and respect your distinction between novelty and significance. However, we would like to clarify that our primary contribution is not merely the proposal of an idea, but its rigorous systematic implementation and empirical demonstration, which transforms a high-level concept into a robust, benchmarked, and reproducible methodology.
>
> Specifically, we contend that **"knowing the direction" is fundamentally different from "making it work."** The blog post you reference provides a conceptual sketch but critically lacks the quantitative results and ablation studies on established benchmarks required to validate the approach for the research community.
>
> Just as the research community spent the period from 2017 to 2020 turning the plausible idea of "using Transformers for vision" (as seen in early works like Image Transformer and Non-local networks) into scalable, state-of-the-art models ViT, our Video-RAG's contribution follows a similar trajectory. We contend that transforming a nascent idea into such an empirically-proven system represents a significant contribution, fully aligned with NeurIPS's standards.
>
> To our knowledge, **our work moves beyond conceptual pointers to deliver the first rigorously validated implementation of integrating multi-modal tools via RAG into video-LVLMs**. By achieving state-of-the-art results on multiple challenging benchmarks (e.g., Video-MME, LongVideoBench), we establish a solid baseline and provide a practical, reproducible framework.
>
> Our visualizations in the main text provide direct qualitative evidence for Video-RAG's quantitative gains. They reveal the underlying mechanism: retrieved text sharpens the LVLM's cross-modal attention on crucial visual details and aligns vision-text semantics, which directly accounts for the performance improvements.
>
> In summary, we respectfully maintain that Video-RAG represents a significant and practical contribution by being the first work to systematically implement and validate this multi-tool RAG approach for video-LVLMs on standard benchmarks. We hope this clarification adequately addresses your remaining concerns and underscores the value of our work to the NeurIPS community.
>
> Thank you again for your time and meticulous review.
>
> Best regards,
>
> Authors

---

### Official Review · Reviewer_BTw3 · 2025-07-02

**Clarity:** 3
**Significance:** 3
**Originality:** 3
**Rating:** 4
**Confidence:** 4

**Summary:**

This paper proposes Video-RAG, a novel training-free and visually-aligned retrieval-augmented pipeline for long video comprehension. Instead of directly increasing the number of sampled frames or relying on proprietary LLMs, the authors leverage open-source tools (e.g., Whisper, EasyOCR, APE) to extract textual information from videos (ASR, OCR, DET). These auxiliary texts are then filtered using RAG mechanisms to select query-relevant information, which is finally integrated with sampled video frames and the user query into a large video-language model (LVLM). The proposed method achieves consistent performance gains across multiple long-video benchmarks (Video-MME, MLVU, LongVideoBench) and even outperforms GPT-4o when integrated with a 72B LVLM.

**Questions:**

1. How robust is Video-RAG to frame sampling errors, especially when key visual cues are missed due to uniform sampling in ultra-long videos?
  2. Could the authors elaborate on how spatial relations (e.g., “on top of”) are computed in the scene graph from raw object coordinates? Why is “Obj Relation” missing in Figure 2 despite an apparent object interaction?
  3. Has the method been evaluated or extended to handle temporally structured queries (e.g., "after X happens, what does Y do")? How would Video-RAG handle such temporal dependencies given its current frame-wise processing?

**Ethical Concerns:**

["NO or VERY MINOR ethics concerns only"]

**Final Justification:**

The proposed method for long video understanding represents a promising direction in lightweight agentic design, eliminating the need for large-scale model training.

**Limitations:**

The method depends on the existing models.

**Quality:**

3

**Strengths And Weaknesses:**

# Strengths
1. Video-RAG introduces a lightweight framework that requires no model training or fine-tuning. It can be directly applied to any LVLM with minimal computation overhead, making it highly practical and easy to adopt in real-world or low-resource scenarios.
2. Unlike prior RAG approaches that treat videos as plain text, this work extracts and retrieves auxiliary texts (OCR, ASR, object detection) that are semantically aligned with the visual content. This alignment significantly improves cross-modality grounding and reduces hallucinations.
3. The method demonstrates solid and consistent improvements across various long video QA benchmarks. When paired with LLaVA-Video 72B, Video-RAG achieves performance surpassing GPT-4o, with only ~2K additional tokens and marginal GPU overhead.

# Weaknesses
1. While Video-RAG avoids increasing the frame count directly, its overall performance remains fundamentally constrained by the initial uniform sampling strategy. If a query-relevant moment (e.g., a keyframe containing a crucial visual cue) is not sampled, the downstream RAG process cannot recover that information, limiting its robustness to extremely long videos.
2. The paper introduces a scene-graph-based preprocessing step to convert raw object detection outputs into structured descriptions, including spatial relations. However, it lacks a clear explanation of how these relations are determined (e.g., what geometric rules define “on top of” or “next to”). Moreover, Figure 2 shows the "Obj Relation" as NULL, even though a visible spatial relation ("books on table") should likely be present, raising concerns about completeness and reliability.
3. Although the method enhances static visual-textual alignment, it does not model temporal dynamics or sequence-level reasoning, which are crucial for many long video tasks (e.g., procedural understanding, action tracking, cause-effect). All auxiliary texts are retrieved independently per frame, without leveraging inter-frame continuity or temporal structure.

---

> ### Author Rebuttal · Authors · 2025-07-29
>
> We sincerely thank you for the constructive and valuable comments. The concerns are addressed as follows.
>
> > **Q1**: Video-RAG's overall performance remains fundamentally constrained by the initial uniform sampling strategy. How robust is Video-RAG to frame sampling errors especially in ultra-long videos?
>
> **A1**: Thanks for your constructive suggestions. We employed **uniform sampling** for two primary reasons:
> - To ensure that our frame sampling strategy aligns with the majority of LVLMs, thereby **demonstrating the plug-and-play capability of our method without reliance on key frames selection**.
> - To facilitate a fair comparison. Empirical evidence indicates that uniform sampling provides the most computationally balanced and generalized solution for video understanding, as corroborated by state-of-the-art open-source models such as Qwen2.5-VL and LLaVA-Video.
>
> In response to your question about how robust Video-RAG is to frame sampling errors, we conducted additional experiments incorporating the ASK[1] (CVPR 2025) keyframe selection method alongside Video-RAG. The results from Video-MME reveal **Video-RAG demonstrates robustness against the loss of keyframe information, particularly in the context of long videos**. This finding underscores two important implications:
> - Negligible orthogonal effects between keyframe selection and Video-RAG.
> - Video-RAG highlights the system's ability to compensate for the absence of critical visual cues through auxiliary text support.
>
> | Method                                          | Short | Medium | Long | Overall|
> | :--------------------------------------         | :-: | :----: | :----: | :----: |
> | Baseline    | 75.4 | 62.6 | 51.8 | 63.3 |
> | Baseline + ASK[1]   | 76.1 | 62.9 | 54.4 | 64.5 |
> | Baselin + Video-RAG   | 77.0 | **69.3** | 63.8 | 70.0 |
> | Baseline + ASK[1] + Video-RAG  | **78.1** | **69.3** | **63.9** | **70.4** |
> *Baseline is the LLaVA-Video-7B with 64 frames for inference.*
>
> > **Q2**: Could the authors elaborate on how spatial relations are computed in the scene graph from raw object coordinates?
>
> **A2**: The spatial relationships are determined by  **calculating the positions of bounding boxes generated through object detection**. The computational workflow is as follows:
>
> ---
>
> **Input**: Accepts a list of objects, each characterized by an ID, class label, and bounding box represented as [xmin, ymin, width, height].
>
> **Processing**:
> 1. Constructs a directed graph using NetworkX, where nodes represent objects and edges represent spatial relationships.
> 2. Performs pure geometric computations (without external dependencies) to determine pairwise relationships between objects:
> - Overlap (determined by bounding box intersection)
> - Relative positioning (including left/right and above/below relationships, where one bounding box is completely situated in the designated direction relative to another without overlap)
> - Prevents duplicate edges by enforcing a node1 < node2 ordering
>
> **Output**: Produces natural language descriptions based on spatial relationships, formatted as follows: "Object 3 (dog) is to the left of Object 7 (cat)."
>
> ---
>
> > **Q3**: Why is “Obj Relation” missing in Figure 2 despite an apparent object interaction?
>
> **A3**: This is because the figure illustrates a reasoning process where **the LVLM determined that addressing this particular question required only positional and number information in the query decouple phase, making object relations unnecessary (hence NULL)**.
>
> We would like to further clarify:
>
> - **Different models may utilize varied reasoning paradigms**. For example, Long-LLaVA-7B would consider object relations to answering the same question.
>
> - **Object relations are fundamentally derived from the positions of objects for better understanding by LVLM**. More advanced models generally depend less on explicit relation annotations, favoring instead a direct analysis that utilizes raw bounding box data.
>
> > **Q4**: Questions about temporal reasoning of Video-RAG.
>
> > **Q4.a**: Although the method enhances static visual-textual alignment, it does not model temporal dynamics or sequence-level reasoning, which are crucial for many long video tasks.
>
> **A4.a**: In long video understanding, modeling temporal dynamics or sequence-level information is indeed crucial. However, we deliberately avoided explicit modeling of these aspects to ensure **improve performance in video understanding across diverse durations while keeping modeling complexity to a minimum**. The temporal modeling techniques you referred to are indeed complementary to our work, and we sincerely appreciate your suggestions for pursuing this avenue in future research.
>
> > **Q4.b**: Has the method been evaluated or extended to handle temporally structured queries?
>
> **A4.b**: An example can be found in Video-MME, which includes temporal reasoning tasks such as "What does Y do after X occurs?" as well as tasks involving the ordering and reasoning of temporal events. In these contexts, **Video-RAG exhibits significant enhancements in both temporal perception and reasoning tasks**.
>
> | Method                                  |  Temporal Perception | Temporal Reasoning |
> | :-------------------------------------- | :----: | :----: |
> | Baseline        | 50.9   | 40.7   |
> | Baseline + Video-RAG   | 69.1   | 45.8   |
> | Performance Gain                    | (+18.2)  | (+5.1)   |
> *Baseline is the Long-LLaVA-7B with 32 frames for inference.*
>
> > **Q4.c**: How would Video-RAG handle such temporal dependencies given its current frame-wise processing?
>
> **A4.c**: The enhancement of temporal perception and reasoning task of Video-RAG arises from the use of temporally structured text encoding, which involves the **chronological integration of auxiliary texts** (such as frame-wise detection transitioning to tracking, incremental ASR facilitating semantic coherence, and sequential OCR reflecting text dynamics). This approach effectively captures video dynamics without requiring explicit temporal modeling.
>
> [1] Tang X, et al. Adaptive keyframe sampling for long video understanding. CVPR 2025.

---

> > ### Comment · Reviewer_BTw3 · 2025-08-07
> >
> > Thank you for the clarification and additional experiments, which have addressed my concerns.

---

> ### Author Response · Authors · 2025-08-05
>
> Dear Reviewer BTw3,
> ﻿
>
> Thanks again for your great efforts and constructive advice in reviewing this paper! As the discussion period progresses, we expect your feedback and thoughts on our reply. We look forward to hearing from you, and we can further address unclear explanations and remaining concerns if any.
> ﻿
>
> Best regards,
> ﻿
>
> Authors

---

> ### Comment · Area_Chair_trg7 · 2025-08-05
> **Requesting Response to Rebuttal**
>
> Dear Reviewer,
>
> please respond to the authors, so that the authors know their rebuttal has been read.
>
> Best regards, AC

---

> ### Author Response · Authors · 2025-08-07
>
> We sincerely thank you again for your review and are happy to see that the concerns you raised have been resolved💗.

---

### Official Review · Reviewer_aLCV · 2025-07-03

**Clarity:** 3
**Significance:** 3
**Originality:** 3
**Rating:** 5
**Confidence:** 5

**Summary:**

This paper proposes Video-RAG, a training-free pipeline to enhance long video understanding for large multimodal models. This approach firstly leverages open-source tools to extract visually-aligned auxiliary texts (e.g., OCR, ASR, object detection) from videos, integrates them via Retrieval-Augmented Generation (RAG) to align cross-modal information, and feeds them into VLMs alongside video frames and queries. Evaluations on benchmarks including Video-MME, MLVU, and LongVideoBench show that Video-RAG improves average performance by 3.2% across six open-source LVLMs, with a 72B model even outperforming proprietary models like GPT-4o. Key contributions include integrating RAG with open-source LVLMs, a plug-and-play design, and achieving proprietary-level performance with open-source models.

**Questions:**

1. How does Video-RAG perform on videos with poor audio quality (e.g., noisy environments) where ASR accuracy drops significantly?
2. What is the impact of using lower-quality open-source tools (e.g., a less accurate OCR than EasyOCR) on Video-RAG’s performance?
3. Are there cases where auxiliary texts introduce noise, and how does the pipeline mitigate such negative effects?

**Ethical Concerns:**

["NO or VERY MINOR ethics concerns only"]

**Final Justification:**

I have carefully read the author's response, which has addressed most of my concerns. I believe this work is practical for long video understanding, and I have no further questions. I tend to accept this paper.

**Limitations:**

1. The novelty of this paper is somewhat limited and this paper focuses on the engineering techniques for video understanding.
2. The proposed method is training-free but this paper lacks further research on how to train video agents with tools.

**Quality:**

3

**Strengths And Weaknesses:**

Strengths:
1. The proposed method is training-free and plug-and-play, which also supports some proprietary models.
2. The proposed method obtains performance improvements on several long video benchmarks, such as VideoMME, MLVU, and LongVideoBench.
3. The proposed method is easy to follow and might perform on downstream applications.

Weaknesses:
1. The performance of Video-RAG is determined by the quality of the external tools (EasyOCR, Whisper, APE). Errors or inaccuracies in transcription, OCR, or object detection will inevitably propagate and potentially mislead the LVLM. The paper does not sufficiently analyze the impact of such errors.
2. The inference latency is higher compared to end-to-end models. The process involves several tools to extract information from long videos which will slow down the inference speed for practical applications.
3. This paper focuses on single-turn process which limits the performance for iterative reasoning or multi-turn conversations.
4. This paper lacks the experimental / qualitative comparisons with previous VideoAgent-like methods, such as VideoAgent[1,2, 3], and other open-source well-established baselines, such as OmAgent for video understanding, which also adopts similar tools, such as ASR.


References\
[1] Wang et al. VideoAgent: Long-form Video Understanding with Large Language Model as Agent. ECCV 2024.\
[2] Fan et al. VideoAgent: A Memory-augmented Multimodal Agent for Video Understanding. ECCV 2024.\
[3] Jeong et al. VideoRAG: Retrieval-Augmented Generation over Video Corpus. ACL 2025.

---

> ### Author Rebuttal · Authors · 2025-07-29
>
> Thank you for the time, thorough comments, and nice suggestions. We are pleased to clarify your questions step-by-step.
>
> > **Q1**: The errors or inaccuracies in transcription, OCR, or object detection that will inevitably propagate and potentially mislead the LVLM.
>
> **A1**: Follow your suggestion, we replaced the components of Video-RAG with alternative components that demonstrated lower performance, and conducted ablation studies on the Video-MME, revealing **only a negligible drop in performance, demonstrating the robustness of Video-RAG to component variability**.
>
> Specifically, we have substituted EasyOCR in the OCR component with Tesseract OCR[1] and the OCRAD API. For the DET component, we have replaced APE with Owl ViT[2] and GRiT[3]. Additionally, in the ASR component, we have replaced Whisper with the SpeechRecognition library and the Vosk API.
>
> | Method                  | Short | Medium | Long | Overall |
> | :---------------------- | :----: | :----: | :----: | :----: |
> | Video-RAG (Base) | 77.0 | **69.3** | 63.8 | **70.0** |
> | replace with Tesseract OCR [1]       | 77.0 | 68.3 | 63.2 | 69.5 |
> | replace with OCRAD API              | **77.6** | 67.8 | 63.9 | 69.7 |
> | replace with Owl ViT [2]             | 76.7 | **69.3** | 63.6 | 69.9 |
> | replace with GRiT [3]                | 76.4 | 68.4 | **64.0** | 69.6 |
> | replace with SpeechRecognition library  | 77.4 | 67.8 | 63.3 | 69.5 |
> | replace with Vosk API               | 76.3 | 68.2 | 63.7 | 69.4 |
> *All models adopt LLaVA-Video-7B with 64 frames for inference.*
>
> > **Q2**: Are there cases where auxiliary texts introduce noise, and how does the pipeline mitigate such negative effects?
>
> **A2**: To investigate this, we injected controlled noise into the text outputs of external tools and evaluated LLaVA-Video-7B-64frames equipped with Video-RAG in Video-MME. Results indicate that **Video-RAG demonstrates robustness in handling noise**. The underlying mechanism is that **the key contextual information retrieved through the RAG process has a high information density**, which helps effectively mitigate performance degradation due to noise and redundant content by filtering out such interference.
>
> | Noise Ratio   | 0% | 10% | 20% | 30% | 40% | 50% |
> | :-------------       | :--: | :--: | :--: | :--: | :--: | :--: |
> | Overall Acc. w/ RAG  | 70.0 | 69.8 | 69.6 | 69.4 | 69.6 | 69.7 |
> | Overall Acc. w/o RAG | 68.6 | 68.3 | 68.2 | 68.0 | 67.9 | 67.6 |
>
> > **Q3**: The inference latency is higher compared to end-to-end models, which slows down the inference speed for practical applications.
>
> **A3**: We appreciate your observation regarding the additional latency introduced by the tools associated with Video-RAG. It’s important to note that our analysis suggests that **a significant portion of the inference latency for long videos is primarily due to the LVLM itself**. The contribution of these tools to the overall inference time decreases notably with the increase in the number of sampled frames. As highlighted in Supplementary Table 3, when processing videos with 384 frames, we draw the following conclusions:
>
> - The base end-to-end LVLM (LongVA-7B [4]) incurs an approximate inference time of 20 seconds per sample on Video-MME.
> - With the cost of additional ~11 seconds inference time increasing, our Video-RAG can achieve a substantial performance improvement of 9.9%.
>
> > **Q4**: This paper focuses on single-turn process which limits the performance for iterative reasoning or multi-turn conversations.
>
> **A4**: Following your suggestion, we have conducted experiments involving Video-MME with IRCoT[5], demonstrating that employing a multi-turn scheme in Video-RAG resulted in a 1.3% enhancement in performance for long videos.
>
> Consequently, we assert that the examination of multi-round question answering warrants further investigation, and we intend to pursue additional research in this domain in the future. Notably, we focused our experiments on long videos (exceeding 30 minutes) because multi-turn question answering is particularly effective for long videos that require complex reasoning.
>
> | Method                                                |  Long Video Acc. |
> | :--------------------------------------               | :------: |
> | Baseline            | 44.1 |
> | Baseline + Video-RAG (1-turn)    | 59.8 |
> | Baseline + Video-RAG (3-turn, IRCoT) | **61.1** |
> *Baseline is the  Long-LLaVA-7B with 32 frames for inference.*
>
> > **Q5**: This paper lacks experimental/qualitative comparisons with previous VideoAgent-like methods.
>
> **A5**: Following your suggestion, we have conducted further comparative experiments involving the VideoAgent-like methods VideoAgent[6], VideoAgent[7], and LLoVi[8]. The results indicate that **Video-RAG consistently outperforms other leading VideoAgent-like methodologies**. Given the considerable API resources required by agent-based approaches, these experiments were executed using our Video-MME-mini benchmark, as detailed in Supplementary Section 3. We would like to clarify that VideoRAG[9] was not initially designed for video understanding tasks, and OmAgent[10] incorporates a highly complex architecture specifically tailored to leverage GPT-4o. We attempted to substitute GPT-4o in OmAgent with LLaVA-Video-72B, but this resulted in inaccurate model outputs. As a result, we were unable to perform comparative experiments on VideoRAG and OmAgent. We intend to incorporate these experimental findings into the main text in future work.
>
> | Method                                    | Short | Medium | Long | Overall |
> | :--------------------------------------   | :-: | :----: | :----: | :----: |
> | Video-RAG (Ours)                     | **85.6** | **74.4** | **68.9** | **76.3** |
> | VideoAgent[6]                        | 50.0 | 46.8 | 42.1 | 46.3 |
> | VideoAgent[7]                        | 64.8 | 56.6 | 44.2 | 55.2 |
> | LLoVi[8]                             | 79.0 | 69.0 | 65.9 | 71.3 |
>
> We use LLaVA-Video-72B in 64 frames as the LVLM in Video-RAG, while other Agent-like methods are all driven by GPT-4.
>
> > **Q6**: The novelty of this paper is somewhat limited and this paper focuses on the engineering techniques for video understanding. The proposed method is training-free but this paper lacks further research on how to train video agents with tools.
>
> **A6**: We appreciate the reviewer's insightful observation. While Video-RAG incorporates some established components, we wish to elucidate its principal contributions as follows:
>
> - **Essence of Video-RAG**: Video-RAG fundamentally establishes a straightforward yet innovative workflow. By leveraging RAG, it assimilates outputs from a variety of multimodal tools, thereby equipping LVLMs with a more comprehensive and information-rich context. This integration significantly ameliorates performance in video understanding tasks.
>
> - **Value proposition of Video-RAG**: The primary significance of this work stems from our recognition that the potential of LVLMs remains underutilized. Video-RAG illustrates that supplying a sufficiently dense and relevant context can more effectively orient the LVLM’s attention towards crucial visual elements. This enhancement promotes improved multimodal interaction and fusion. Future research could expand on Video-RAG by integrating additional external tools, facilitating adaptations to diverse application scenarios.
>
> - **Broader research relevance**: It is important to highlight that the design of such workflows that integrate tool outputs for targeted tasks or the distillation of high-quality contextual inputs represents a significant research avenue within the broader LLM domain. Recent methodologies, including Agentic-like approaches, context compression techniques, and context state management strategies, fall within this thematic area. Our analysis reveals that papers fitting this category comprise approximately 10%–12% of the accepted submissions at NeurIPS 2024, emphasizing the critical importance of this area of inquiry.
>
> - **Future directions regarding training**: We acknowledge the reviewer’s pertinent observations concerning the significance of training methodologies, such as recent works (e.g., ToolRL[11]) that utilize reinforcement learning strategies to enhance the tool-calling capabilities of LLMs. Your suggestion provides a valuable trajectory for future optimization: investigating how to further fine-tune LVLMs so they can more adeptly incorporate high-quality contextual information sourced from external tools.
>
> [1] R. Smith. An Overview of the Tesseract OCR Engine. Ninth International Conference on Document Analysis and Recognition. 2007.
>
> [2] M Minderer, et al. Simple Open-Vocabulary Object Detection with Vision Transformers. ECCV 2022.
>
> [3] Jialian Wu, et al. GRiT: A Generative Region-to-text Transformer for Object Understanding. ECCV 2024.
>
> [4] Peiyuan Zhang, et al. Long Context Transfer from Language to Vision.
>
> [5] Trivedi H, et al. Interleaving retrieval with chain-of-thought reasoning for knowledge-intensive multi-step questions. ACL 2023.
>
> [6] Fan, et al. VideoAgent: A Memory-augmented Multimodal Agent for Video Understanding. ECCV 2024.
>
> [7] Wang, et al. VideoAgent: Long-form Video Understanding with Large Language Model as Agent. ECCV 2024.
>
> [8] Ce Zhang, et al. A Simple LLM Framework for Long-Range Video Question-Answering. EMNLP 2024.
>
> [9] Jeong, et al. VideoRAG: Retrieval-Augmented Generation over Video Corpus. ACL 2025.
>
> [10] Lu Zhang, et al. OmAgent: A Multi-modal Agent Framework for Complex Video Understanding with Task Divide-and-Conquer.
>
> [11] Cheng Qian, et al. ToolRL: Reward is All Tool Learning Needs.

---

> ### Author Response · Authors · 2025-08-05
>
> Dear Reviewer aLCV,
> ﻿
>
> Thanks again for your great efforts and constructive advice in reviewing this paper! As the discussion period progresses, we expect your feedback and thoughts on our reply. We look forward to hearing from you, and we can further address unclear explanations and remaining concerns if any.
> ﻿
>
> Best regards,
> ﻿
>
> Authors

---

> ### Comment · Area_Chair_trg7 · 2025-08-05
> **Requesting Response to Rebuttal**
>
> Dear Reviewer,
>
> please respond to the authors, so that the authors know their rebuttal has been read.
>
> Best regards, AC

---

### Comment · Area_Chair_trg7 · 2025-08-02
**Discussion Phase**

Dear Authors and Reviewers,

I would like to thank the authors for providing detailed rebuttal messages.

Dear **reviewers**, I would like to encourage you to carefully read all other reviews and the author responses and engage in an open exchange with the authors. Please post your first response as soon as possible within the discussion time window, so there is time for back and forth discussion with the authors. Ideally, all reviewers will respond to the authors, so that the authors know their rebuttal has been read.

Best regards, AC

---

### Author Response · Authors · 2025-08-08
**General Response**

We sincerely thank all reviewers for your detailed and valuable comments. All reviewers (**aLCV**, **BTw3**, **YJ27**, **LHvP**) acknowledged the effectiveness of Video-RAG, noting consistent performance improvements across long-video benchmarks such as VideoMME, MLVU, and LongVideoBench. The training-free and plug-and-play design was highlighted for enabling easy integration with various LVLMs, including proprietary models (**aLCV**, **BTw3**). The use of semantically aligned auxiliary texts beyond frames improves cross-modal interaction and reduces hallucinations (**BTw3**, **LHvP**). The clarity of presentation, reliability of results, and thoroughness of experiments were also praised (**YJ27**, **LHvP**), with the approach demonstrating strong real-world applicability (**aLCV**, **BTw3**).

Based on these comments, we conclude some noteworthy replies for the reviewers, including:

- **[Reviewer aLCV, LHvP]** We have validated the robustness of Video-RAG to component errors and noisy auxiliary texts through ablation and controlled noise experiments.
- **[Reviewer aLCV]** We clarify that the added inference latency is marginal compared to the LVLM’s cost, and the substantial performance gain justifies the overhead. (see Table 3 in the supplementary)
- **[Reviewer aLCV]** We have extended Video-RAG to multi-turn reasoning and compared it with existing VideoAgent-like methods, demonstrating its effectiveness and generality.
- **[Reviewer aLCV, YJ27]** We have clarified the novelty of Video-RAG as a training-free, context-augmentation framework that improves LVLMs.
- **[Reviewer BTw3, LHvP]** We clarify that Video-RAG differs from relevance-based frame sampling methods in motivation, methodology, and effectiveness, and does not rely on such techniques for performance gains.
- **[Reviewer BTw3]** We clarify that spatial relations are computed geometrically from object bounding boxes and only used when the LVLM determines them necessary for answering the query.
- **[Reviewer BTw3]** We have demonstrated that Video-RAG enhances temporal perception and reasoning without requiring explicit temporal modeling.
- **[Reviewer YJ27]** We have provided empirical evidence of synergistic effects among components.
- **[Reviewer YJ27]** We have conducted an error analysis on Video-MME, identifying key frame selection and complex reasoning as current bottlenecks, and proposed directions for future improvement.
- **[Reviewer LHvP]** We have shown that Query Decouple is robust to noise and has minimal impact on performance, even with low-accuracy or suboptimal models.

---

### Decision · Program_Chairs · 2025-09-17

**Decision:**

Accept (poster)

**Comment:**

The paper discusses a training-free pipeline that augments LVLMs for long-video QA by extracting visually aligned auxiliary texts, filtering them via RAG, and concatenating them with frames and the query. It reports gains on several benchmarks.

The reviewers found the method practical, clearly presented, and broadly effective across benchmarks. The ablations support robustness to weaker tools and show the auxiliary-text pathway is necessary. The rebuttal clarified spatial relations handling and temporal reasoning improvements.

Some reviews saw the novelty/significance as limited. Further concerns were that the approach uses uniform frame sampling and adds latency and that comparisons to other systems were not fully integrated in the main paper.

During the discussion some of the reviewers' concerns were addressed, e.g. through added analyses and multi-turn comparisons.

Overall, I recommend to **accept** this paper to NeurIPS. Addressing a current problem with a practical method, the paper is very suitable for presentation in my opinion.

---
Request for the final version: It looks like some concurrent work has appeared since the manuscript was created. For example, the below two works seem relevant from a first look, and there may be more. (I do not have any relation to any of the authors or their institutions, this is just the based on an online search.) I would like to ask the authors of this paper to update their related work and references to include more current references for the camera ready version, where appropriate.
- [VideoRAG: Retrieval-Augmented Generation over Video Corpus](https://arxiv.org/abs/2501.05874)
- [VideoRAG: Retrieval-Augmented Generation with Extreme Long-Context Videos](https://arxiv.org/abs/2502.01549)